# Conformations of a highly expressed Z19 α-zein studied with AlphaFold2 and MD simulations

**Niels Johan Christensen**[ORCID]*

Department of Chemistry, University of Copenhagen, Frederiksberg, Denmark

* njc@chem.ku.dk

**Data Availability Statement:** All MD trajectories are available in the same reduced form as used for analysis herein. They can be downloaded in PDB format from Kaggle: https://www.kaggle.com/dsv/7620451

## Abstract

α-zeins are amphiphilic maize seed storage proteins with material properties suitable for a multitude of applications e.g., in renewable plastics, foods, therapeutics and additive manufacturing (3D-printing). To exploit their full potential, molecular-level insights are essential. The difficulties in experimental atomic-resolution characterization of α-zeins have resulted in a diversity of published molecular models. However, deep-learning α-zein models are largely unexplored. Therefore, this work studies an AlphaFold2 (AF2) model of a highly expressed α-zein using molecular dynamics (MD) simulations. The sequence of the α-zein cZ19C2 gave a loosely packed AF2 model with 7 α-helical segments connected by turns/loops. Compact tertiary structure was limited to a C-terminal bundle of three α-helices, each showing notable agreement with a published consensus sequence. Aiming to chart possible α-zein conformations in practically relevant solvents, rather than the native solid-state, the AF2 model was subjected to MD simulations in water/ethanol mixtures with varying ethanol concentrations. Despite giving structurally diverse endpoints, the simulations showed several patterns: In water and low ethanol concentrations, the model rapidly formed compact globular structures, largely preserving the C-terminal bundle. At $\geq$ 50 mol% ethanol, extended conformations prevailed, consistent with previous SAXS studies. Tertiary structure was partially stabilized in water and low ethanol concentrations, but was disrupted in $\geq$ 50 mol% ethanol. Aggregated results indicated minor increases in helicity with ethanol concentration. β-sheet content was consistently low ($\sim$ 1%) across all conditions. Beyond structural dynamics, the rapid formation of branched α-zein aggregates in aqueous environments was highlighted. Furthermore, aqueous simulations revealed favorable interactions between the protein and the crosslinking agent glycidyl methacrylate (GMA). The proximity of GMA epoxide carbons and side chain hydroxyl oxygens simultaneously suggested accessible reactive sites in compact α-zein conformations and pre-reaction geometries for methacrylation. The findings may assist in expanding the applications of these technologically significant proteins, e.g., by guiding chemical modifications.

**Funding:** This work was supported by a research grant 40932 from VILLUM FONDEN (www.veluxfoundations.dk) awarded to NJC. The funders had no role in study design, data collection and analysis, decision to publish, or preparation of the manuscript.

**Competing interests:** The author has declared that no competing interests exist.

## Introduction

The growing awareness of the harmful effects of fossil-derived plastics on the Earth's ecosystem has sparked a quest for new biomaterials as sustainable alternatives [1]. Agriculture is one source of such materials, as biopolymers from crops favourably balance accessibility, biocompatibility, and sustainability [2]. Moreover, the physicochemical properties of biopolymers can often be tailored to extend their range of applications [3].

Zeins are examples of biopolymers with desirable material characteristics such as hydrophobicity, thermoplasticity and the ability to form non-covalent viscoelastic networks [4]. They are maize grain storage proteins (prolamins) comprising more than half of the $\sim 10\%$ protein content of maize seeds [5] where they are deposited inside endosperm cells in membrane-enveloped organelles known as protein bodies [6]. There are four major zein groups (α, β, γ, δ) differing by hydrophobicity and apparent molecular weights in gel experiments. α-zein is the most hydrophobic and abundant of the groups, comprising 75–85% of the total zein [7].

The properties of zeins have been harnessed in a wide range of applications, such as edible coatings [8], medicinal biomaterials [9], thermoresponsive gels [10], nanoparticles with tuneable morphologies [11], and nanocarriers for hydrophobic drugs [12]. In the field of additive manufacturing (3D-printing) the thermoplasticity of zein has been exploited in fused deposition modelling (FDM) where zein is melted and extruded through a nozzle [13, 14]. Additionally, ethanolic zein solutions have been used for 3D bioprinting [15] and for 4D printing in drug release applications [16]. The tailorability of zein, as evidenced in various studies, has facilitated its broader application. For example, brittleness in microfluidic systems made from zein was reduced by oleic acid and monoglyceride as plasticizers [17]. Reduced zein brittleness and enhanced yield strength have also been achieved by anti-solvent precipitation [18]. Hydration, adhesion, and extensibility have been improved by oxidation with hydrogen peroxide and horseradish peroxidase [19] and by cold plasma treatment [20]. The mechanical properties of zein have been engineered by crosslinking [21, 22]. Recently, excessive aggregation in 4D-printing with zein was prevented by tuning the water content of an ethanol/water supporting bath to 40–75% [16].

The α-zein fraction has been extensively studied. Early gel experiments found that it comprises two major α-zein size classes with apparent molecular weights of 19 kDa and 22 kDa, leading to their respective names Z19 and Z22 [23]. It was later discovered that their actual molecular weights were higher, but the names persisted [7]. The first complete zein sequence determined was a Z19 zein obtained in 1981 by Geraghty *et al.* by reverse transcription of a protein body polysome zein mRNA [24]. This protein sequence named A30 (UniProtKB ID: P02859) displayed characteristics later found in all α-zeins. Notably, there was an abundance of dipeptide repeats such as Gln-Gln and the last two-thirds of the protein consisted of 7–8 repeats of a degenerate $\sim 20$ amino acid repeat unit. Similar repeats were later found in the Z19 zein A20 (UniProtKB ID: P04703) and the Z22 zein B49 (UniProtKB ID: P05815). These repeats were represented by the consensus sequence NPAAYLQQQQLLPFNQLA(V/A)(L/A) [25]. Parallel efforts by Pedersen *et al.* gave the sequence of the Z19-zein ZG99, alternatively named gZ19AB1, with UniProtKB ID: P04704 [26]. This sequence, in conjunction with the sequence of Z22-zein cZ22B1 (UniProtKB ID: P04698) [27], was used to propose the consensus sequence LQQ(F/L)LP(A/F)NQL(A/L)(A/V)ANSPAYLQQ, which formed the basis of a widely referenced structural model of zein [28] described below. Collectively, the studies by Marks *et al.* [29] and Woo *et al.* [30] summarizes major α-zein protein sequences, as outlined in S1 Appendix. The topological tree in Fig 1 shows how these sequences fall into the three subfamilies named "19 kD B", "19 kD D", and "22 kD" [30]. Importantly, these studies concur that while there are on the order of 100 zein genes [31] only a subset is detectably expressed in the

common inbred maize varieties W64A [29] and B73 [30]. In the latter study, only 8 α-zein genes were substantially expressed. The top-three expressed random selected clones measured by percentage expressed sequence tags (ESTs) were αz19B1, αz19B3, and αz22z1, see Fig 1. The second-highest expressed clone, αz19B3, corresponds to the cZ19C2 zein examined herein.

Every α-zein protein sequence can be divided into four primary sections: a signal peptide, an N-terminal turn, a variable number of approximately 20-residue homologous repeat units, and a C-terminal turn. This is illustrated in Fig 2 using zein gZ19AB1 as an example [28]. In this article, the numbering of residues does not include the signal peptide.

There is no general agreement on the definition of the repeat unit and published numbers range from 6 to 9 repeats depending on definition and zein sequence [35]. It has been proposed that Z22 zeins are larger than Z19 zeins due to exactly one extra repeat [35]. However, regardless of repeat definition, they constitute about two-thirds of any α-zein sequence and must largely define the proteins' spatial structure.

Due to the challenges of zein separation [36] most experimental characterizations have been carried out on mixtures of α-zeins in alcoholic solution [37, 38] or in the solid state [39, 40] However, two studies on Z19 zeins offer relevant context for the present work. In the first study, Forato *et al.* [41] studied α-zein extracted from the maize variety BR451 which lacks Z22 expression. Gel experiments showed solely Z19 zeins which were characterized without further purification or sequencing. From solid state FTIR the authors reported α-helix segmentation consistent with $\sim 20$ residue repeats and a secondary structure distribution of 46% α-helix, 22% β-sheet, 23% turns, and 13% undetermined structure. An extended structure was inferred from $^1$H NMR (70% deuterated EtOH) and SAXS (90% EtOH, 3.6 mg zein /mL). The authors proposed a hairpin structure based on αz19D2 (UniProt ID: Q946V7) containing helical segments,

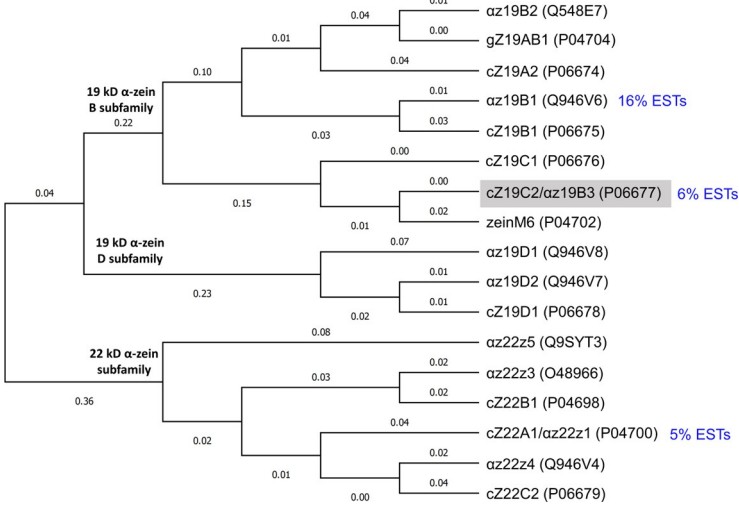

**Fig 1. Topological tree for α-zein protein sequences.** Zein names are from the studies [23, 29, 30]. UniProtKB IDs are given in brackets. Expressed sequence tag (EST) percentages are given for the three most expressed randomly chosen clones in B73 endosperm [30]. The cZ19C2 zein studied here is indicated by a grey background. The tree excludes signal peptides. The tree was built with MEGA11 [32] using a multiple sequence alignment of the mature α-zein sequences carried out with the integrated CLUSTAL W [33]. The Maximum Likelihood method and the JTT matrix-based model were used to infer the evolutionary history [34]. The tree with the highest log likelihood (-2686.46) is shown. The initial tree(s) for the heuristic search were automatically obtained using the Neighbor-Join and BioNJ algorithms on a pairwise distance matrix estimated with the JTT model. The topology with the highest log likelihood value was selected. This analysis included 17 amino acid sequences, with a total of 276 positions in the final dataset. Branch lines do not scale with branch lengths; however, the branch lengths, measured in the number of substitutions per site, are displayed next to the branches.

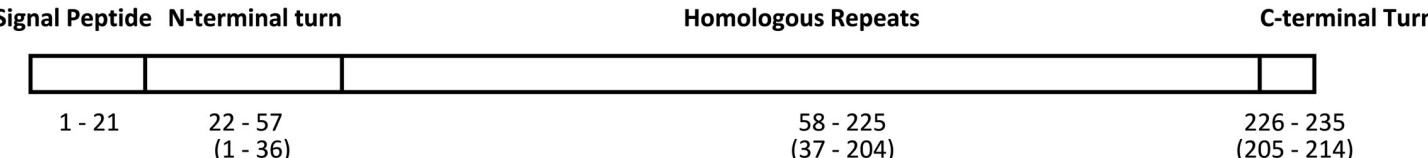

**Fig 2. Overall α-zein sequence organization.** As an example, the segmentation of gZ19AB1 [28] is indicated by residue numbers. Numbers in brackets exclude the signal peptide.

sheets, and turns with dimensions of 12 Å × 120–130 Å. In the other Z19-focused study, Cabra *et al.* [36] purified a Z19 α-zein from waxy yellow dent corn seeds to give a single MALDI--TOF-MS peak of 24.535 kDa. By sequencing the first 20 amino acids (TIFPQCSQAPIASLLP-PYLP) they identified the protein as zein M6 [23] (UniProtKB ID: P04702) using BLAST. The CD secondary structure distribution was 40.0% α-helix, 19.5% β-sheet, 15.4% coils, and 25.1% undetermined secondary structure for the isolated zein (0.08–0.25 mg/mL) in 70% (v/v) methanol. CD tertiary structure prediction suggested that the α-helices folded into unspecified compact structures. This differs from the extended structure assignments from SAXS and NMR studies on other α-zein preparations in alcoholic solution [38, 41–43].

Interestingly, a contemporary BLAST search on the 20 first amino acids reported by Cabra *et al.* returned three additional sequence hits from UniProtKB/SwissProt (updated 2022/10/30 https://blast.ncbi.nlm.nih.gov/) including cZ19C2, see S36 Fig in S1 File. As the calculated molecular weight of cZ19C2 (24.087 kDa) approximates the molecular weight from MALDI--TOF-MS, the spectroscopic findings by Cabra *et al.* may possibly apply to the zein studied here.

Despite decades of research, no experimental atomic-level structures of zeins have been published. Instead, the current understanding of zein molecular structure relies on low-resolution experimental methods such as SAXS and a multitude of structural models. Early idealized "consensus-sequence models" leveraged the α-zein repeat structure: A single ∼ 20 residue consensus sequence represents all repeats and is modelled as a single α-helix, except for terminal residues that reside in turns connecting to neighbor helices. Different spatial arrangements of multiple identical consensus helices give distinct models. The consensus sequence by Argos *et al.* mentioned above has been used in several published models, including their own 1983 model [28]. The model is a hollow cylinder formed by lateral association of nine anti-parallel consensus-helices linked end-to-end by glutamine-repeat loops. The glutamine side chains enable hydrogen bonds in the axial direction between zein monomers, while lateral interactions between helices stabilize the monomer and enable lateral self-assembly of zein monomers. The 1993 model by Garratt *et al.* [35] for Z22 zeins also assumes lateral association of antiparallel consensus α-helices, but on a two-dimensional hexagonal grid. This grid model can grow by association of additional monomers while minimizing voids, to emulate α-zein packing in protein bodies. Finally, the 1997 consensus model by Matsushima *et al.* [43] consists of antiparallel repeat helices laterally stacked perpendicular to the helical axis, giving a linear ribbon-like model with dimensions 130 Å × 12 Å × 30 Å. The model was used to explain both limited aggregation in SAXS experiments (see below) and extensive aggregation assumed in protein bodies.

After the consensus models, all-atom models incorporating complete sequence information have become prevalent. A widely cited example is the 2005 model by Momany *et al.* [44] for the cZ19C2 α-zein. The authors combined human intuition and molecular modelling to build the α-zein monomer as a superhelix consisting of three axially stacked heterotrimeric coiled coils, respectively containing the residue ranges (37–54, 55–77, 78–92), (93–112, 113–127, 128–146), and (147–166, 167–185, 186–204). Each coiled coil was stabilized by nonpolar

interactions with a molecule of the pigment lutein. In MD simulations, the model was structurally stable in a mixture of 90% methanol and 10% water, but degraded in pure water.

More recently, there has been an increase in automatically generated all-atom α-zein models from homology modelling (e.g., [45–47]) and *ab initio* modelling (e.g., [48, 49]). These studies moved beyond static single structures by subjecting the models to MD simulations.

Recent applications of MD simulations directly targeting α-zein structural understanding are exemplified by the research by Erickson *et al.* [50, 51]. They undertook [50] comparative MD studies of zein M6 (UniProtKB: P04702) constructed both as the superhelical model by Momany *et al.* [44] and as the extended hairpin molecular model by Forato *et al.* [41] in both 75% and 45% (v/v) ethanol/water. Only the hairpin model agreed with their CD-studies on commercial zein mixtures by giving increased β-sheet in more hydrophilic solvent. The study [51] focused on β-sheet propensities of peptide segments from the three most expressed α-zein clones αz19b1, αz19b3 (= cZ19C2) and αz22z1 found by Woo *et al*. The cZ19C2 peptides were from positions 9–19, 21–28, 75–90 and 148–157. The 21–28 peptide had the largest β-sheet propensity ($\sim$ 30–45%) whereas the other peptides had a maximum β-sheet content of $\sim$ 10%.

In summary, the large body of modelling studies outlined above delineates three main groups of α-zein monomer models: consensus-sequence models, manually built atomistic models, and automatically generated atomistic models (including homology and ab initio models). While all models have importantly shaped the thinking about α-zein structure, it is useful to consider some of their possible limitations. A potential caveat for models in the first two groups is their reliance on human judgment for establishing a consensus sequence and/or assigning secondary and tertiary structure. Furthermore, in case of consensus models, their inherent neglect of variations in repeat sequences renders them unable to capture any conformational diversity that may arise from such heterogeneity. The automatically generated atomistic models also have limitations. The lack of templates for homology modelling of α-zeins can manifest as generation of widely different models from identical sequences. This is illustrated in Fig A in S2 Appendix in S1 File for the zein examined herein. Ab initio models do not have this issue, but they are susceptible to inadequate sampling or theoretical treatment.

The present approach does not eliminate all of these concerns and could potentially introduce new issues. Nonetheless, as the modelling route differs from previous research this study contributes by offering a new perspective. The modelling herein draws inspiration from the large conformational variability of α-zein under different environmental conditions indicated by numerous spectroscopic studies. This implies that a single native structure, if such exists, is unlikely to represent the diverse array of α-zein conformations in practical applications. Therefore, the main goal of the present study is not to suggest a single native-like state of zein, but rather to contribute to charting the conformational landscape of α-zein in various solution environments. The environments addressed here are ethanol, water and their binary mixtures, as the latter are common solution systems [52, 53]. The set of 0, 2, 23, 50, and 100 mol% ethanol concentrations in water were selected to encompass simulations with ethanol percentages typical of experimental solution systems, i.e., 60–95 vol% ethanol [52] as well as concentrations approaching the solubility limits of α-zein (i.e., approximating pure water or pure ethanol). The latter concentrations, representing extreme conditions, were included with the expectation that they might perturb the protein structure, thus extending the types of conformations sampled. The findings may serve as a groundwork for a comprehensive mapping of α-zein conformations across a broad range of conditions.

The deep-learning method AlphaFold2 [54] was used to build the α-zein structure. AlphaFold2 is not a homology- or an *ab initio* method, but may be seen as an extension of both approaches, expanding their capabilities through deep-learning integration of evolutionary, physical, and geometric constraints of known proteins. It was hypothesized that this level of

data integration might partially compensate for the lack of homologous experimental structure templates for α-zein. All-atom MD simulations in varied ethanol/water solvents were used to sample the conformations of the AlphaFold2 model. In this context, a particular benefit of the model was its loose arrangement of secondary structure elements. This allowed large-scale rearrangements during MD without bias towards a densely packed tertiary structure as often found in homology- or *ab initio* α-zein models. Illustrating the latter, *ab initio* models generated from the same sequence (Fig B in S2 Appendix in S1 File) show a dense packing that may hinder conformational sampling without obviously ensuring a more realistic initial state.

## Methods

### AlphaFold2 model

The protein sequence for the α-zein cZ19C2 [29] excluding signal peptide (residues 1–21) was obtained via the accession number P06677 from UniProtKB [55]. The sequence was submitted via the ColabFold [56] web-interface to generate AlphaFold2 [54] models. The best ranked model was used for all further studies. The protonation state of the model was assigned with the GROMACS pdb2gmx program, giving a net charge of +1. The assigned protonation states of ionizable groups are given in Table A in S3 Appendix in S1 File. To ensure adequate solvation, the principal molecular axes of the model were aligned with the Cartesian axes in VMD [57] prior to solvation.

### GROMACS all-atom simulations

Solvent boxes containing of 0, 2, 23, 50, and 100 mol% ethanol in water were made by substituting water for the required number of ethanol molecules in a pre-equilibrated pure ethanol box (ethanol_T298.15.pdb.gz from https://virtualchemistry.org/ff/GAFF-ESP-2012.zip) using GROMACS tools. The resulting simulation box compositions are shown in Table 1. Each box was subjected to energy minimization followed by MD equilibration (100 ps NVT and 40.1 ns NPT at T = 300 K) to give stable densities. The equilibrated solvent boxes were used to solvate the AlphaFold2 model to give cubic protein-solvent boxes with at least 15 Å of solvent on each side of the protein. The +1 charge of the protein was neutralized by substitution of a randomly chosen solvent molecule with a single chloride ion. Each protein-solvent box corresponding to a particular ethanol concentration was subjected to the GROMACS simulation protocol outlined below.

The GROMACS all-atom simulation protocol consisted of (1) energy minimization to relax steric clashes using a target maximum force of 1000 kJ mol$^{-1}$ nm$^{-1}$, (2) 100 ps MD equilibration with protein heavy atom position restraints in the NVT ensemble at T = 300 K using the velocity rescaling (V-rescale) thermostat, (3) 100 ps MD equilibration in the NPT ensemble (P = 1 bar, T = 300 K) with protein heavy atom position restraints using the Parrinello-Rahman

**Table 1. System composition for GROMACS all-atom simulations.**

| System | Ethanol molecules | Water molecules | Cl$^-$ |
|---|---|---|---|
| 100 mol% water | 0 | 57033 | 1 |
| 2 mol% ethanol | 985 | 54796 | 1 |
| 23 mol% ethanol | 8788 | 30174 | 1 |
| 50 mol% ethanol | 16050 | 16301 | 1 |
| 100 mol% ethanol | 17728 | 0 | 1 |
| 100 mol% water (ff99SB*-ILDN) | 0 | 57033 | 1 |
| 24 mol% water (ff99SB*-ILDN) | 8919 | 28902 | 1 |

barostat pressure coupling, (4) production MD run in the NPT ensemble (P = 1 bar, T = 300 K). The leap-frog algorithm was used to integrate the equations of motion using a timestep of 2 fs and holonomic constraints were imposed with LINCS [58]. Periodic boundary conditions and Particle Mesh Ewald long-range electrostatics were used with cubic interpolation and a Fourier spacing of 0.16 nm. System densities after 100 ps NPT equilibration (step 3) are listed in Table B in S3 Appendix in S1 File.

For the 400 ns simulations with the OPLS-AA/L force field, three simulation replicates labelled Seed 1, Seed 2, and Seed 3, were carried out for each of the 5 ethanol concentrations giving a total of 15 × 400 ns MD simulations. Each simulation replicate differed by initial atomic velocity randomization to improve sampling, The simulations were carried out with GROMACS 2022 [59] using the OPLS-AA/L force field [60] for protein and the SPC/E water model [61]. GAFF [62] parameters for ethanol were obtained from virtualchemistry.org as ethanol.itp from the package https://virtualchemistry.org/ff/GAFF-ESP-2012.zip.

To probe the behaviour of the α-zein model with a newer force field, four independently prepared 400 ns control simulations (2 seeds for 100 mol% water, and 2 seeds for 24 mol% ethanol) were carried out in GROMACS with the TIP3P water model [63] and the ff99SB*-ILDN force field for protein [64, 65]. The parameters were imported from (https://github.com/bestlab/force_fields/blob/master/gromacs_format/README). The 100 mol% water (seed 1) simulation was extended to 800 ns. Time series for these supplementary simulations are given in S27-S34 Figs in S1 File

## Additional equilibration of a globular α-zein model

Based on its globularity and structural stability in a long MD trajectory, the 1 μs all-atom GROMACS simulation endpoint in water (seed 3) was chosen as a starting point for multiple zein simulations. The structure was further equilibrated in a water box in Desmond [66, 67] for 500 ns in the NPT ensemble at 300 K and 1 bar with default simulation protocol settings, using the OPLS3 [68] force field for protein and the TIP3P water model.

## All-atom simulations of multiple α-zeins

The Maestro [69] graphical user interface and Desmond was used for preparing and executing all-atom simulation of the multi-zein system. In Maestro, a non-interacting configuration of six copies of the globular α-zein was constructed by placing the monomer structures at positions corresponding to midpoints of the faces of a sufficiently sized virtual cube, see Fig 16A. This structure was solvated in a cubic box, giving a total of 489201 atoms (156161 water molecules) followed by a 500 ns MD simulation in Desmond with the standard pre-simulation relaxation protocol and NPT simulation settings. The OPLS3 force field was used for the protein and the TIP3P [63] model for water.

## Coarse-grained simulations of multiple α-zeins

Coarse-grained simulations on multiple copies of the globular α-zein conformation were carried out in GROMACS with the Martini 2.2 force field [70]. The all-atom α-zein monomer was converted to coarse-grained coordinates and topology using the Martini script Martinize. py. The -dssp option was used with the script, providing secondary structure assignment using DSSP-based [71] analysis of backbone dihedrals and hydrogen bonds. The structure before and after coarse-graining is shown in Fig 17A and 17B, respectively. To build a crowded model system mimicking a pre-aggregation state, 73 copies of the coarse-grained α-zein monomer were randomly inserted into a $250 \times 250 \times 250$ Å$^3$ box, with a minimum van der Waals distance of 4 Å between proteins. The box was solvated with a pre-equilibrated

$160 \times 160 \times 160$ Å$^3$ box of 35937 coarse-grained Martini water molecules, using a minimum van der Waals distance of 2.1 Å between waters. The net positive charge of +73 was neutralized by automatically replacing 73 randomly selected waters with chloride ions. After energy minimization, the system in Fig 17C was subjected to three NPT (P = 1 bar, T = 303 K) simulations differing by initial velocity randomization, that were each run for 375 ns (15000000 steps of 25 fs). The simulation endpoints are shown in Fig 17D–17I.

### Interactions with a crosslinking reagent

The 1 μs all-atom GROMACS simulation endpoint in water (seed 3) was used to probe interactions with the crosslinking reagent glycidyl methacrylate (GMA). GMA was built and minimized in Maestro, and a spherical distribution of 48 GMA molecules was set up around the α-zein with 47 Å between the centroids of each GMA molecule and the α-zein centroid. This ensured a minimum initial distance of 7 Å between any GMA molecule and the α-zein surface. This configuration was solvated in a cubic box of 61370 TIP3P molecules giving a buffer of 10 Å on all sides of the protein-GMA configuration. The system was subjected to the default multistage minimization and NPT simulation protocol in Desmond for 257 ns.

### MD analysis and visualization

Prior to analysis, the GROMACS MD trajectories were downsampled to give a time interval of 1 ns between each frame. Molecular figures were made with VMD [57] and Maestro. VMD was used for MD trajectory analysis, including calculation of backbone root mean square deviation (RMSD), solve-accessible surface area (SASA), radius of gyration ($R_g$), and secondary structure based on STRIDE [72].

## Results and discussion

### α-zein sequence and initial structural model

Although all sequences in Table A in S1 Appendix in S1 File are candidates for investigation, here the practical choice was made to study the Z19 α-zein cZ19C2 also known as αz19b3 (UniProtKB ID: P06677). This decision was motivated by several factors, including the expected prevalence of this zein in actual preparations. As noted above, it ranked among the top three zeins in terms of expression level in common maize lines. It is also one of only two zeins currently registered in UniProtKB with protein-level evidence, see Table C in S1 Appendix in S1 File. Furthermore, the relatedness with other zeins, as depicted in Fig 1, suggests that insights from cZ19C2 could potentially apply to other zeins as well. Moreover, the sequence has been cited in several studies, e.g., [73–76], and was used for the atomistic model by Momany *et al.* [44] mentioned above. Finally, cZ19C2 is homologous or possibly identical with the purified α-zein in Cabra *et al.*'s study [36] which makes it a relevant reference point and enables comparison with previous research. In Fig 3A the sequence has been manually aligned with the consensus sequence from Geraghty *et al.* [25], yielding 12 segments each containing up to 20 residues. Towards the C-terminal, this qualitative alignment highlights an increasingly direct mapping between the aligned segments and individual C-terminal helices in the AlphaFold2 model, as discussed in a later section.

Initially, an attempt was made to predict regions with self-assembly potential within the mature cZ19C2 protein sequence. This was motivated by a desire to increase the transparency of the modelling process. Furthermore, it was of interest to see if coiled-coil interactions of the type central to the cZ19C2 model by Momany *et al.* [44] could be detected with contemporary prediction algorithms. Waggawagga [77] and Deepcoil2 [78] were used to scan the sequence

for single-α helix (SAH) domains, canonical heptad-repeat coiled-coil domains, and non-canonical coiled coil domains. However, none of these domains were identified. The study moved forward with prediction of the structure using AlphaFold2.

The AlphaFold2 model is shown in Fig 3B. Its loosely arranged structure superficially resembles models from published experimental α-zein characterizations in alcohol/water mixtures, such as Forato *et al.*'s [41] segmented multi-helix model and Tatham *et al.*'s elongated few-helix model [42]. Overall low predicted lDDT [79] scores (S1 Fig in S1 File) imply high model uncertainty across the sequence, reflecting the absence of α-zein-like structures in the AlphaFold2 training data. The model lacks β-sheets, and has an α-helical content of 68% distributed among seven antiparallel and mostly loosely packed α-helical segments (denoted helix I to VII) of varying length connected by turns or loops. The α-helical segments are initiated by Pro-Ala or, in one case, Arg-Ala. Some conspicuous features of the model include the two central long α-helices (helix II and III), and the bundle of three relatively tightly packed α-helices near the C-terminal (helix IV, V, and VI) which have slightly higher confidence scores, see S1 Fig in S1 File. To further examine the apparently more structurally well-defined C-terminal part of the protein, the full-sequence AlphaFold2 model was compared with partial-sequence AlphaFold2 models built from N89-F219 (segments 6 to 12 in Fig 3A) and N144-T203 (the C-terminal three-helix bundle). As shown in S4 Appendix, structural superposition revealed the α-helical bundle to be highly similar across the three models, indicating internal consistency in the AlphaFold2 predictions.

The α-helices II and III appear unusually long given their lack of stabilizing interactions with adjacent structure [80, 81]. Interestingly, the *ab initio* method C-QUARK also predicted a long helix for a subset (S56 –A86) of the helix II residues, see Fig B in S2 Appendix in S1 File.

A detailed description of the AlphaFold2 model follows. Starting from the C-terminal, the unstructured tail residues Q210 –F219 are preceded by the short helix VII (P205 –Y209). The preceding residues N144 to D204 span the bundle of the three antiparallel α-helices, helix IV (P145 –A163), helix V (R165 –A182), and helix VI (P185 –T203) that are laterally associated with each other, forming a mainly hydrophobic core. Interestingly, the sequence of each helix corresponds to segments 9, 10, and 11 of the zein sequence in the alignment to the consensus sequence by Geraghty *et al.* [25], see Fig 3A. Thus Gln-Gln repeats are inside helices and not in turns linking the helices, in contrast to the consensus model by Argos *et al.* [28]. Instead, N144, N164 and N184 reside in the turns and expose their side chains, possibly enabling hydrogen bonding. The C-terminal α-helical bundle is connected to the long helix III (P110-L143) through a hairpin turn containing N144. Comparison of helix III to the consensus sequence alignment in Fig 3A shows that it spans segments 7 and 8. It could have been reasonable to anticipate a break in helix III at the position corresponding to the start of segment 8, in light of the one-to-one segment-to-α-helix mapping for the C-terminal helices IV, V, and VI. However, in contrast to the latter, gaps were used in segments 7 and 8 for alignment with the consensus sequence. The gaps imply that segment 8 lacks residues corresponding to positions 1–3 in the consensus sequence. Since these consensus sequence positions in segments 9–11 coincide with the start of helices IV - VI, their lack in segment 8 may explain why AlphaFold2 instead merges segments 7 and 8 into the single long α-helix III. Indeed, all α-helices in the AlphaFold2 model are either initiated by Pro-Ala or Arg-Ala, whereas the first two amino acids in segment 8 are Ala-Tyr. Helix III connects via the disordered residues Q98-N109 to the other long helix, helix II (P57-Q97). Unlike helix III, the terminal residues of helix II do not correspond to segment start and stops in the consensus sequence alignment (Fig 3A). Instead, helix II begins in the middle of segment 4, spans segment 5 and terminates in the middle of segment 6. Except for segment 6, these segments share little homology with the consensus sequence. This reflects a gradual loss of recognizable repeat structure towards the α-zein N-

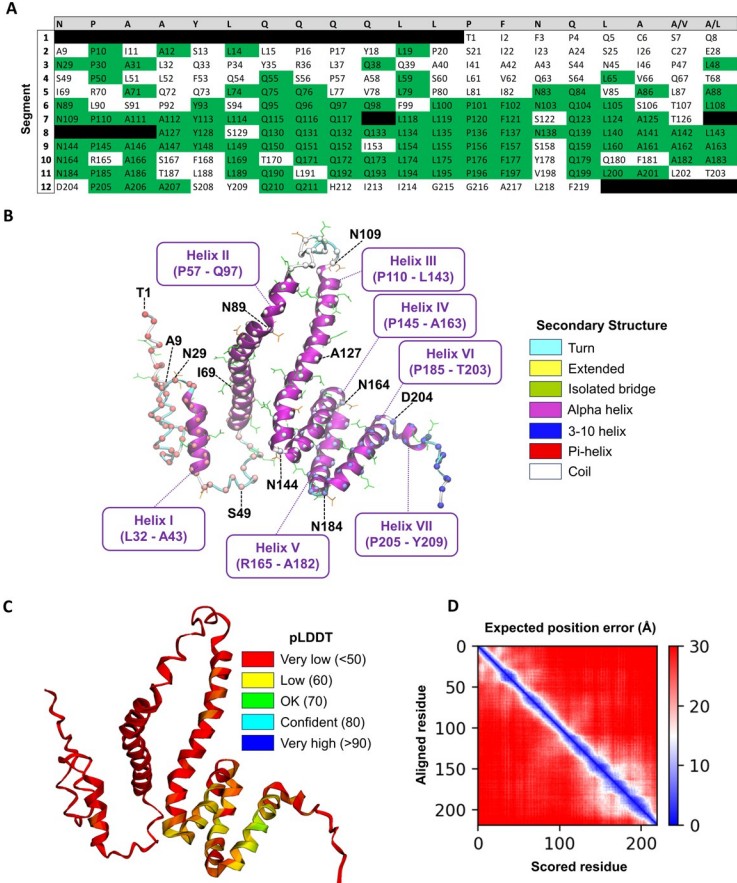

**Fig 3. Consensus sequence comparison and AlphaFold2 model.** A: Qualitative consensus sequence alignment. Top row (grey background) is the consensus repeat sequence by Geraghty *et al.*[25] The other rows contain the mature protein sequence of cZ19C2 (UniProtKB ID: P06677) manually aligned with the consensus repeat sequence in segments of up to 20 amino acids. Green background indicates amino acids matching the consensus sequence. Black background indicates gaps. B: AlphaFold2 model. Residues corresponding to the first amino acid of each row in (A) are indicated by black labels. The α-helices I–VII are annotated in purple. Gln and Asn are shown in green and orange stick representation, respectively. The protein backbone is shown as ribbons coloured by secondary structure. α-carbons are shown as spheres coloured by sequence position (red: N-terminal, blue: C-terminal). C: pLDDT scores indicated by ribbon colour. D: Predicted aligned error (PAE) matrix.

terminus. Helix II is linked via the unstructured residues S44-S56 to the short helix I (L32-A43) which again is linked to the N-terminal tail residues T1 - A31 which exhibit less ordered structure consisting of coils and turns. This region contains the only two cysteines, C6 and C27, which are in close proximity with a S-S distance of 3.4 Å.

## 400 ns All-atom MD simulations

The structural snapshots in Figs 4–8 represent the behavior of the AlphaFold2 model during the fifteen 400 ns MD simulations. In each figure, each row corresponds to a simulation replicate differing by the seed (seed 1, seed 2, seed 3) used for randomizing initial atom velocities. The first column (subfigs A, D, E) shows the initial AlphaFold2 structure as reference. The middle column (subfigs B, E, H) shows 10 superimposed snapshots from the last 100 ns of each simulation, to illustrate conformational variation. The last column (subfigs C, F, I) displays the final structural snapshots at 400 ns.

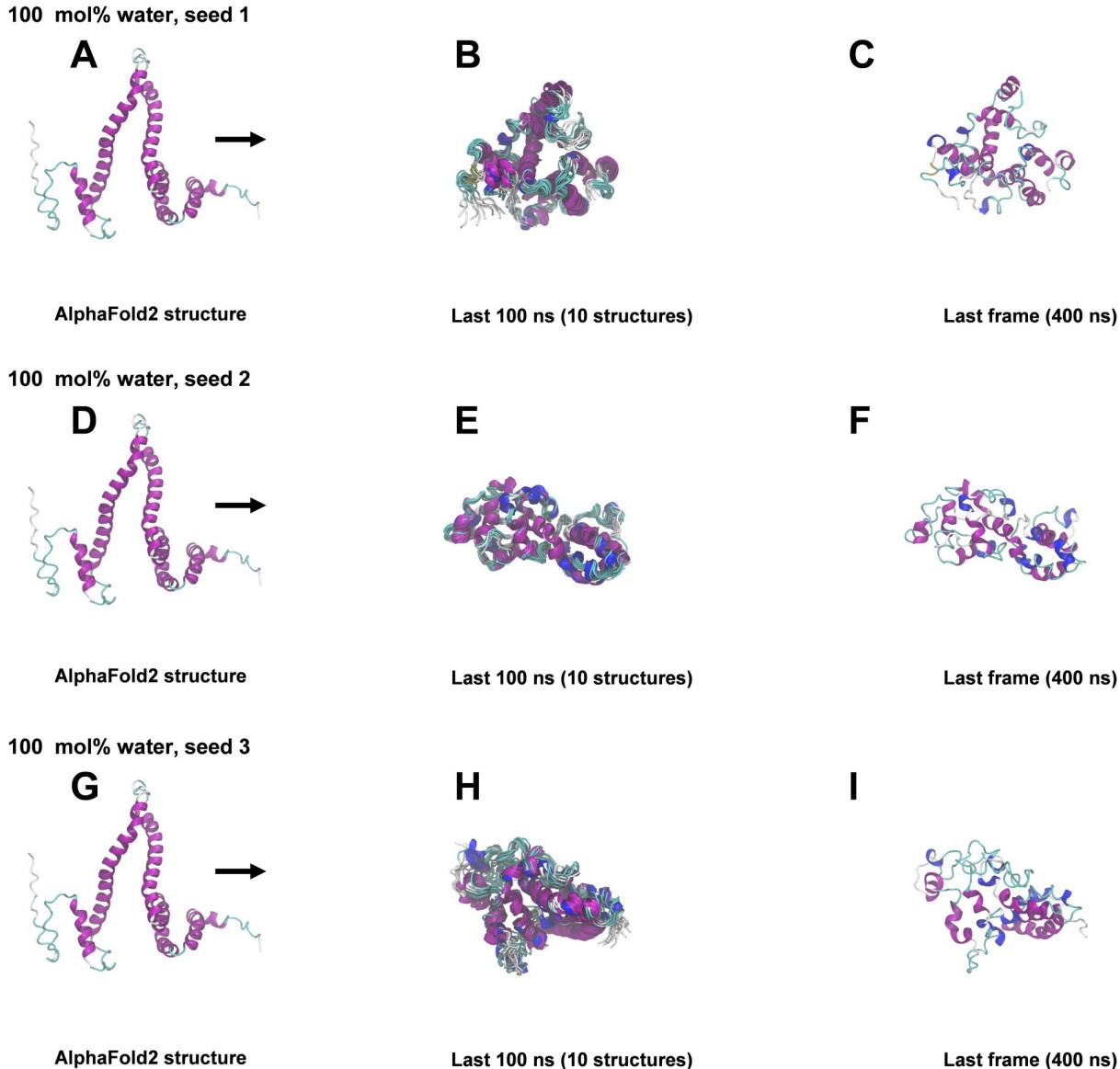

**Fig 4. Snapshots from 400 ns MD simulations in 100 mol% water.** A, D, G: AlphaFold2 structure before simulation. B, E, H: Superimposition of 10 snapshots from the last 100 ns of each simulation. C, F, I: Snapshots at 400 ns. The ribbon representation is colored by secondary structure (see legend in Fig 3B).

The associated time series for backbone root mean square deviation from the AlphaFold2 structure (RMSD), radius of gyration ($R_g$) and protein solvent accessible surface area (SASA) are included in S3 Fig in S1 File. Average values for these time series are shown in Fig 9.

Simulations differing only by the randomization seed used for initial velocities gave distinct structures at 400 ns, as evident from the last columns of Figs 4–8. The divergence between trajectories differing by seed was substantial already early in the simulations, see S5 Appendix. This agrees with the expected sensitivity to perturbations in the loosely packed AlphaFold2 structure. The two long α-helices (helix II and III) did not maintain their initial forms in any of the simulations, but rearranged into various mixtures of secondary structure elements. In both water and in 2 mol% ethanol, the model rapidly collapsed into compact globular forms

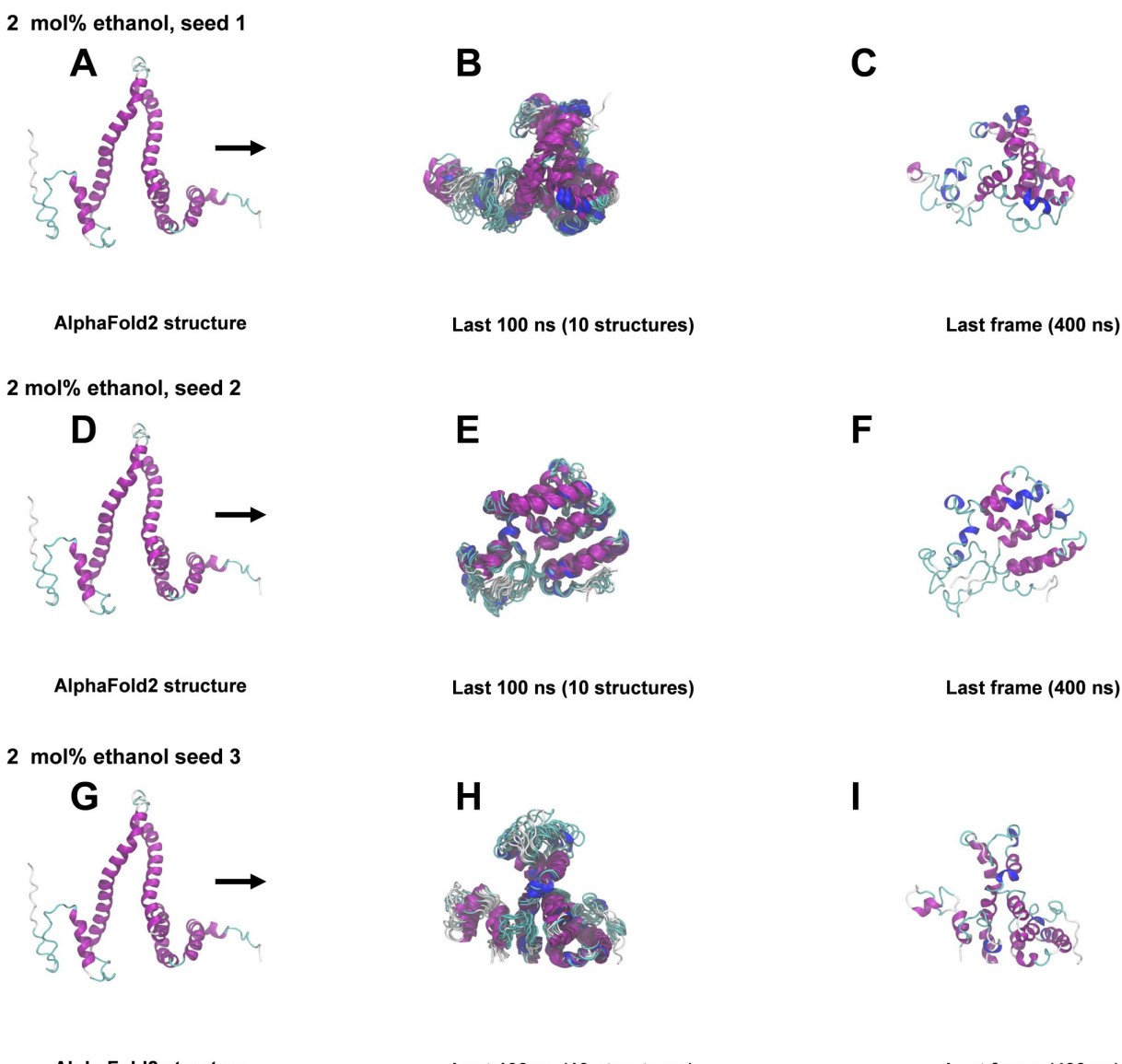

**Fig 5. Snapshots from 400 ns MD simulations in 2 mol% ethanol.** A, D, G: AlphaFold2 structure before simulation. B, E, H: Superimposition of 10 snapshots from the last 100 ns of each simulation. C, F, I: Snapshots at 400 ns. The ribbon representation is colored by secondary structure (see legend in Fig 3B).

that were largely retained for the remainder of the simulations. This is reflected in the fast convergence to relatively low average values in the RMSD, $R_g$ and SASA time series (S3 Fig in S1 File) and by the similarity of superimposed MD snapshots within each simulation (Fig 4 and S3B, S3E, S3H Fig in S1 File). The simulation endpoints (Figs 4 and 5C, 5F and 5I) show the distinct compact structures. With increasing ethanol concentration, a wider range of α-zein conformations were sampled. The increased sampling was subtle at 2 mol% ethanol, but became prominent at 23 mol% ethanol and above. This is seen from the larger fluctuations and higher average values in the RMSD, Rg, and SASA time series (S3 Fig in S1 File) and by less congruent superimposed MD snapshots at higher ethanol concentrations, see Figs 4–8, subfigs B, E, and H. Relative to water simulations, simulations with added ethanol showed

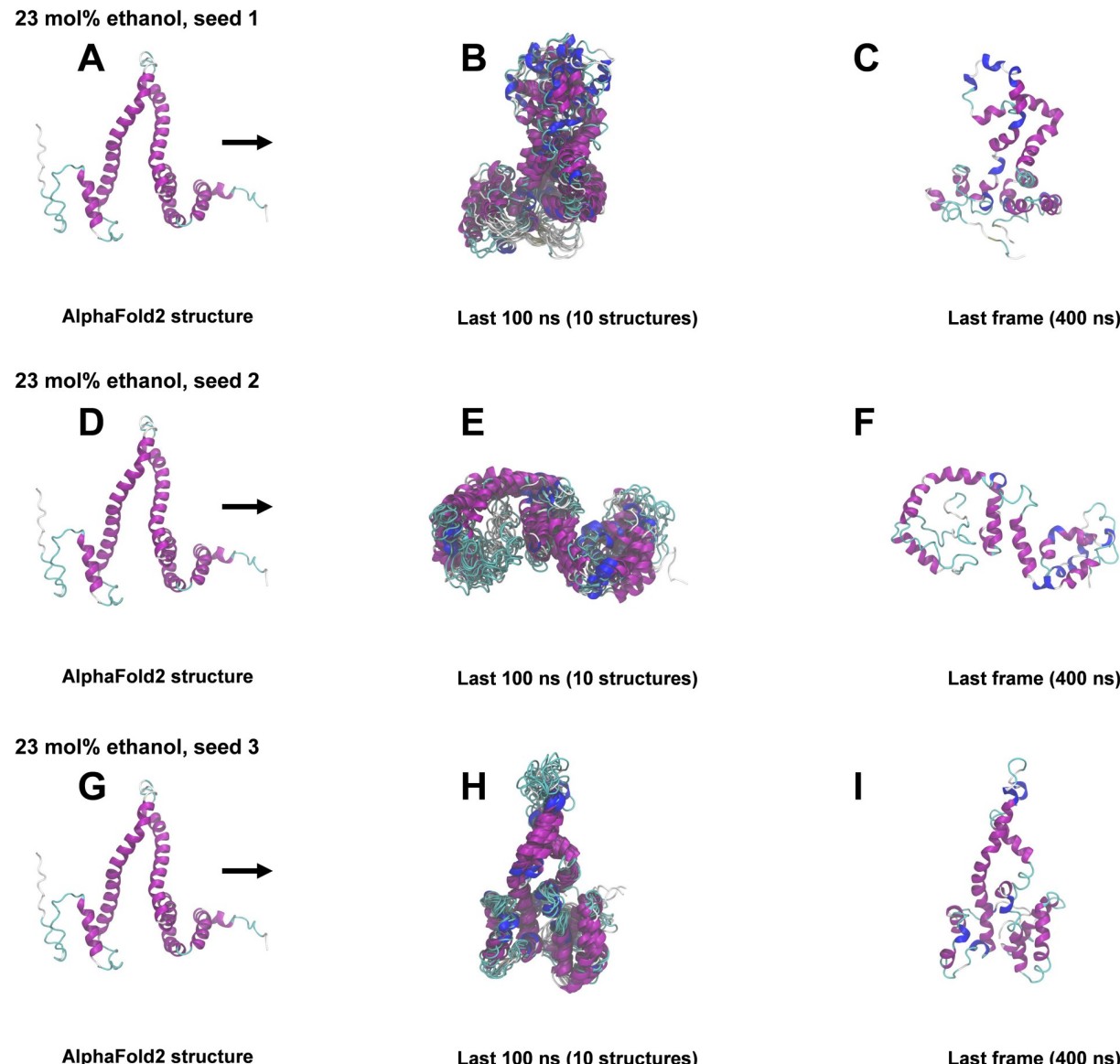

**Fig 6. Snapshots from 400 ns MD simulations in 23 mol% ethanol.** A, D, G: AlphaFold2 structure before simulation. B, E, H: Superimposition of 10 snapshots from the last 100 ns of each simulation. C, F, I: Snapshots at 400 ns. The ribbon representation is colored by secondary structure (see legend in Fig 3B).

structural expansion. This is evident from the time series averages of radius of gyration (Rg) and SASA, calculated across the three simulation seeds for each ethanol concentration, as depicted in Fig 9. For instance, the average $R_g$ almost doubled from water (18.9 Å) to ethanol (36.1 Å). Also, the average SASA in water (12486 Å$^2$) and ethanol (19857 Å$^2$) implies a $\sim 59\%$ SASA increase in ethanol. At high ethanol concentration, these changes were accompanied by disruption of interactions between secondary structure elements (i.e., tertiary structure). This disruption was particularly evident for the elongated structures observed at 50 mol% ethanol (seed 2) and 100 mol% ethanol (seed 1), see Figs 7F and 8C. At 50 mol% ethanol, the three 400 ns simulation replicates sampled expanded α-zein conformations, as indicated by the time series for $R_g$ and SASA (S3 Fig), with one replicate (seed 2) showing pronounced elongation,

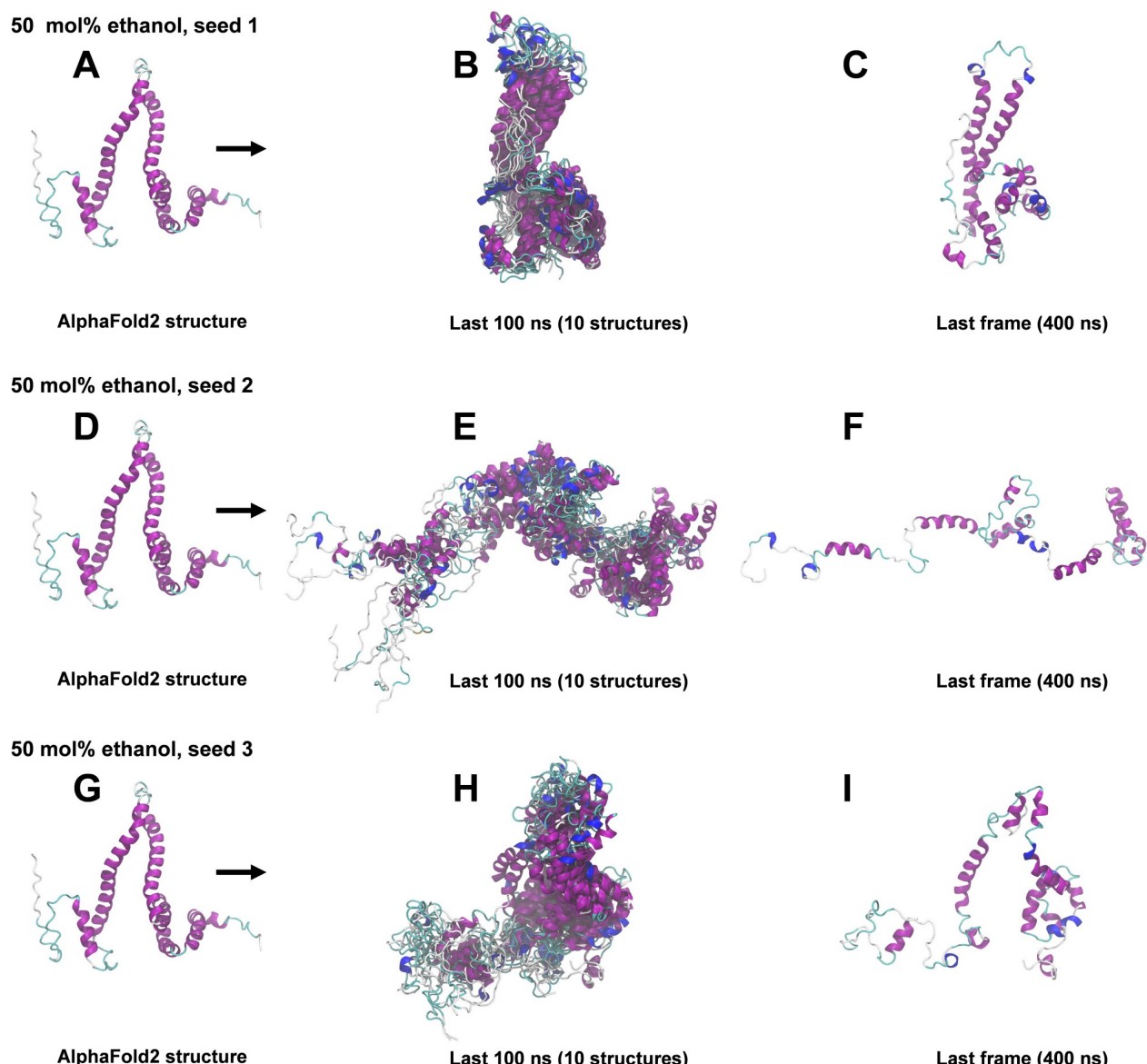

**Fig 7. Snapshots from 400 ns MD simulations in 50 mol% ethanol.** A, D, G: AlphaFold2 structure before simulation. B, E, H: Superimposition of 10 snapshots from the last 100 ns of each simulation. C, F, I: Snapshots at 400 ns. The ribbon representation is colored by secondary structure (see legend in Fig 3B).

see Fig 7F. The onset and extent of expansion varied across the replicates, occurring at the simulation start for seed 2 but only after $\sim 140$ ns for seed 1, underlining the stochastic nature of MD simulations. This highlights the significance of using multiple seeds and conducting long simulations to explore conformational changes. At 100 mol% ethanol, the time series (S3 Fig in S1 File) and structural snapshots resemble the results at 50 mol%. Thus, there was once again an increased exploration of expanded conformations, as reflected by disordered superposition of snapshots. Also, one simulation (seed 1) produced very elongated conformations. The SASA averaged over seeds (19857 Å$^2$) was similar to the 50 mol% ethanol simulation (19599 Å$^2$), while the average radius of gyration (36.1 Å) was significantly larger compared to the 50 mol% simulation (31.8 Å). Nevertheless, the general resemblances between the

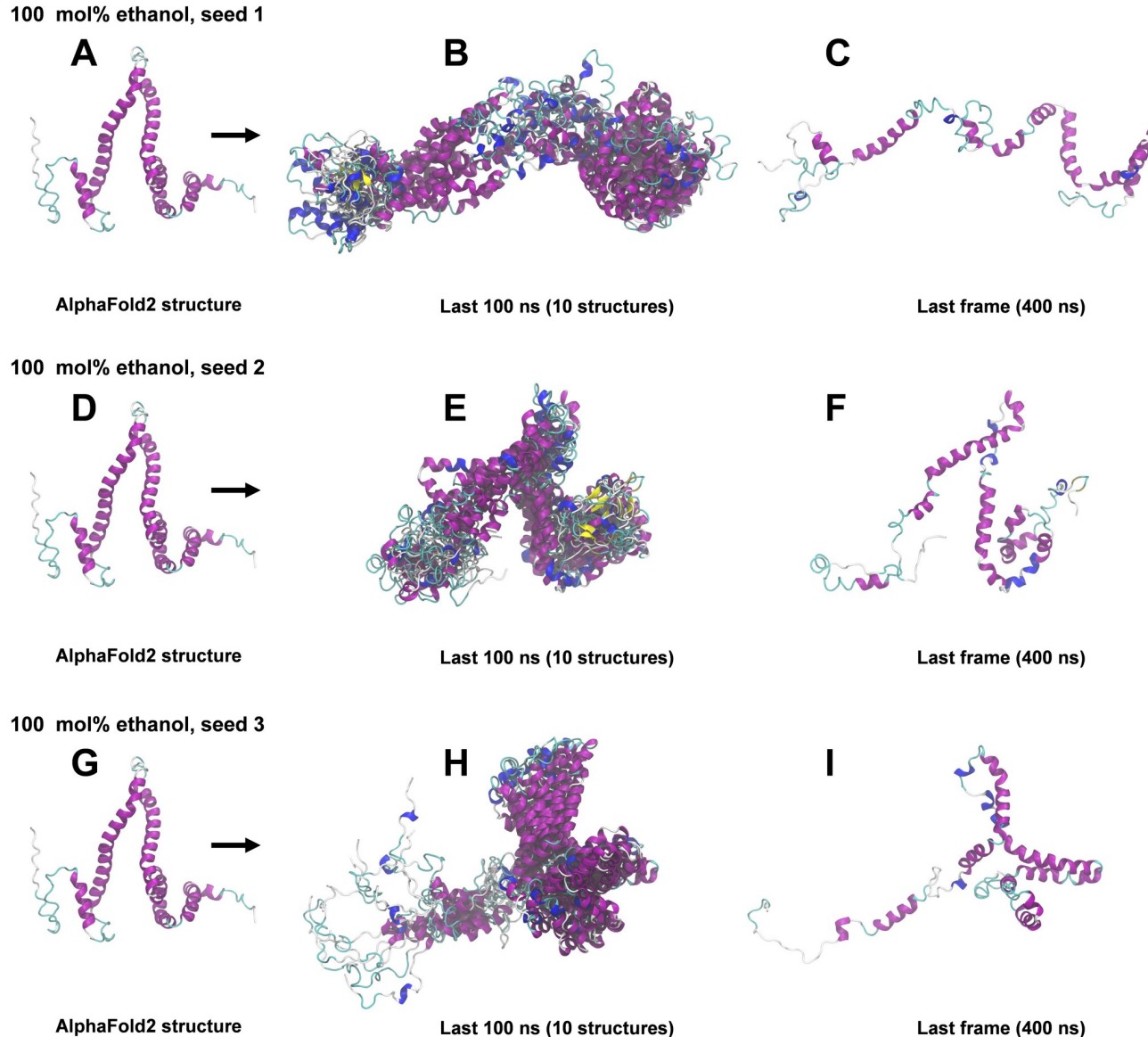

**Fig 8. Snapshots from 400 ns MD simulations in 100 mol% ethanol.** A, D, G: AlphaFold2 structure before simulation. B, E, H: Superimposition of 10 snapshots from the last 100 ns of each 400 ns simulation. C, F, I: Snapshots at 400 ns. The ribbon representation is colored by secondary structure (see legend in Fig 3B).

simulations at 100 mol% ethanol and 50 mol% ethanol may indicate that the impact of ethanol on conformational sampling has reached a plateau at around 50 mol%. A possible concern arising from the pronounced structural expansion at high ethanol concentration ($\geq$ 50 mol%) is that the very elongated zein conformations in some simulations could lead to non-physical interactions between α-zein and its own image in neighboring simulation cells. Thus, these limiting cases should be interpreted with some care. Nevertheless, the elongated conformations are consistent with previous SAXS experiments, as shown in a later section.

Identification of least-mobile protein substructures using MDLovoFit [82] offers an alternative view of how solvent composition influences the α-zein model. MDLovoFit maps rigid and mobile regions in a MD trajectory by aligning progressively larger fractions of Cα-atoms

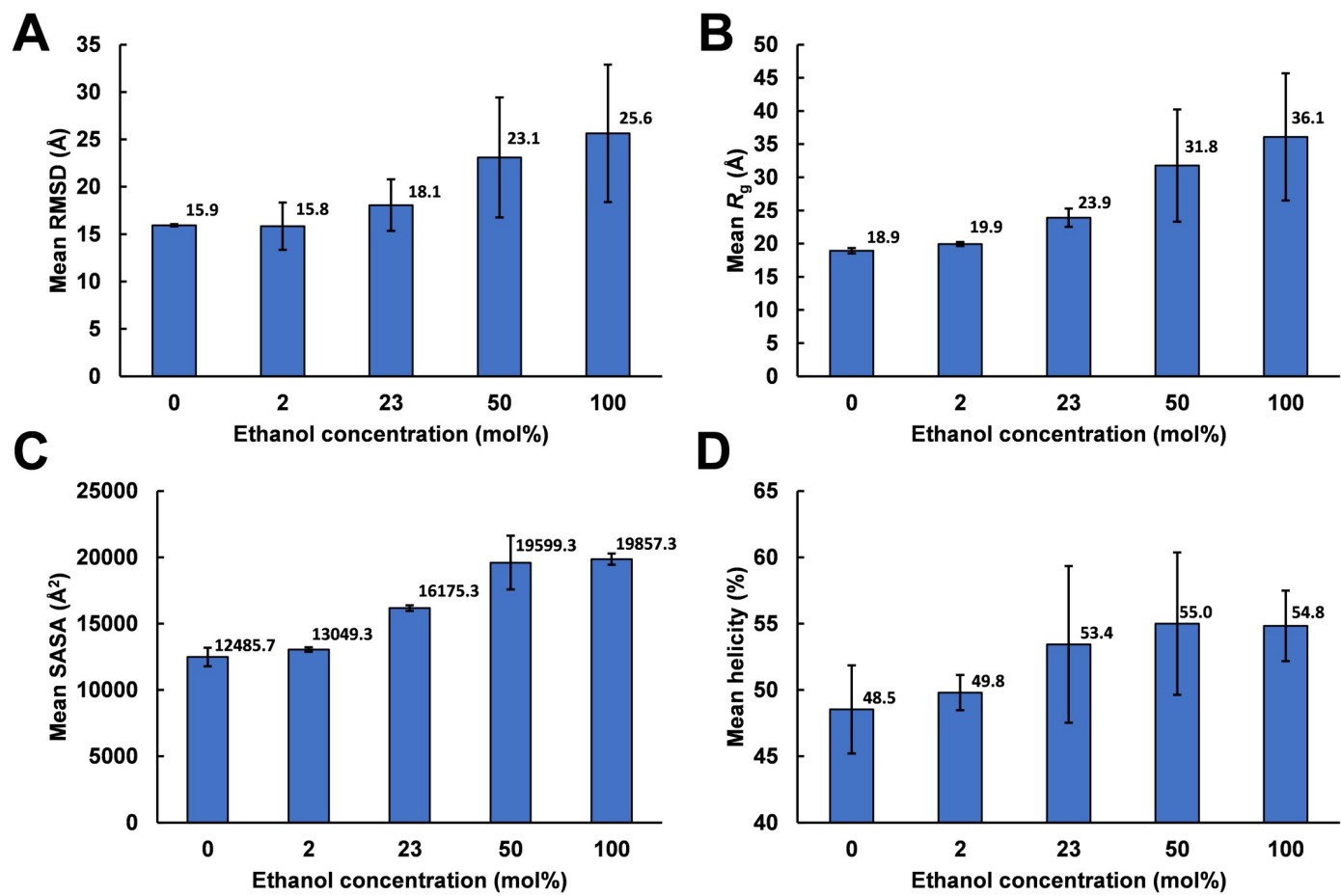

**Fig 9. Time series averages for 400 ns MD simulations.** A: Backbone RMSD relative to the AlphaFold2 structure. B: Radius of gyration ($R_g$). C: Solvent accessible surface area (SASA). D: Total helicity (sum of α- and $3_{10}$-helicity). Values for each ethanol concentration were averaged over the last 100 ns of each simulation time series and then over the three simulations (seed 1–3) per concentration. Error bars indicate standard deviation across seeds seed 1–3. For the associated time series see S3 Fig in S1 File.

(denoted by $\phi$) across the trajectory and computing the RMSD of the aligned atoms. The method was applied to the final 100 ns of each 400 ns MD trajectory. The resulting plot in Fig 10 depicts RMSD for Cα-atom superposition versus the fraction of the total number of Cα atoms ($\phi$).

Each fraction corresponds to the lowest attainable RMSD for all possible subsets of size $\phi$. The plots show that $\sim 30\%$ of the Cα-atoms ($\phi = 0.3$) can be superimposed within a RMSD of 1 Å in the trajectories of all 100 mol% water simulations, all 2 mol% ethanol simulations, and two 23 mol% ethanol simulations (seed 1 and 3). In contrast, only around 10% of the Cα-atoms ($\phi = 0.1$) could be superimposed within 1 Å for one 23 mol% ethanol simulation (seed 2), two 50 mol% ethanol simulations (seed 1 and 3), and one 100 mol% ethanol simulation (seed 3). The extensive conformational variation in the two remaining simulations, 100 mol% ethanol (seed 1) and 50 mol% ethanol (seed 2), implied that only a negligible fraction of Cα atoms could be superimposed within $\sim 1$ Å RMSD.

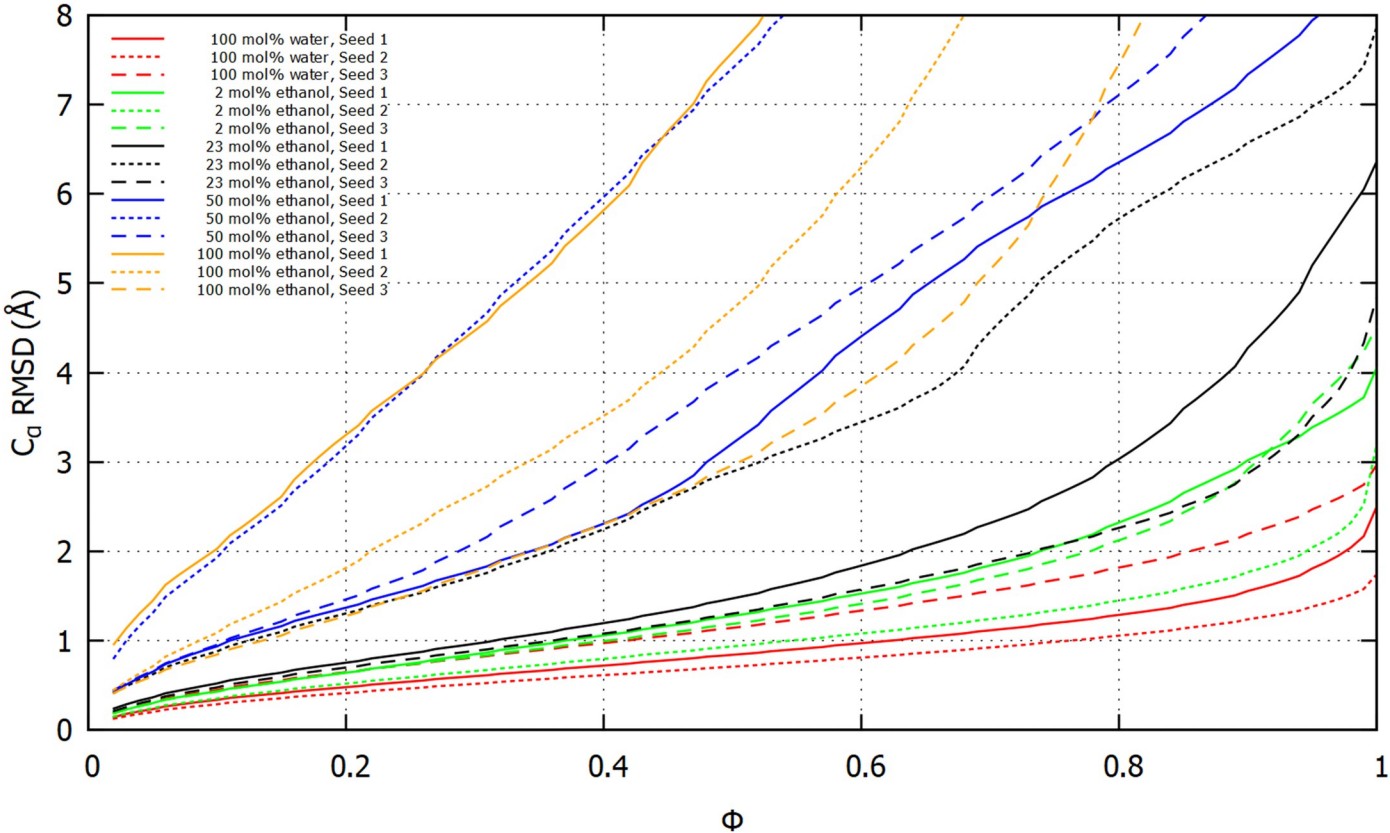

**Fig 10. Least-mobile substructures in the 400 ns MD simulations.** The graphs for each simulation show the lowest RMSD of Cα-atoms after superpositions of all subsets of size $\phi$ (fraction of the total number of Cα atoms). The analysis was carried out on the last 100 ns of each simulation with MDLovoFit [82].

## Helicity

An averaged representation of helicity in all $15 \times 400$ ns simulations is shown in Fig 11. For each residue, the sum of α-helicity and $3_{10}$-helicity averaged over the last 100 ns of all $15 \times 400$

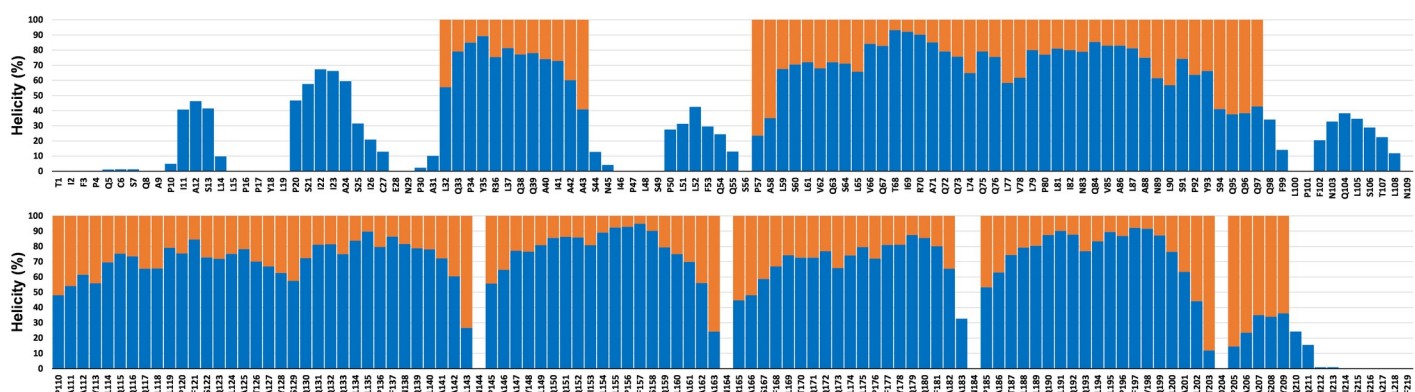

**Fig 11. Helicity per residue.** The height of the blue bars in the foreground indicates the helicity in percent from MD simulations (sum of $3_{10}$ and α-helicity of the last 100 ns of each 400 ns simulation averaged over all $15 \times 400$ ns MD simulations.) The uniform orange bars in the background indicate positions of α-helical residues in the AlphaFold2 model.

ns simulations is plotted with blue bars. The initial AlphaFold2 α-helicity is shown as orange bars. Individual plots for each ethanol concentration are provided in S9-S13 Figs in S1 File.

The plot in Fig 11 shows a substantial retention of helical structure from the AlphaFold2 model.

Moreover, some simulations display helicity in regions that were initially non-helical in AlphaFold2 model (e.g., P20 - C27). Furthermore, specific residues never became helical. Notable examples are N144, N164, and N184, which are located in the turns connecting helix III and IV, helix IV and V, and helix V and VI, respectively. Non-helicity for these residues is readily explained for simulations with high water content. In those cases, stabilization of helix-helix interactions in the C-terminal α-helical-bundle restrained N144, N164, and N184 to reside near their initial positions in the turns connecting the helices. Interestingly, these residues remained non-helical even at high ethanol concentration where helix-helix interactions were destabilized, permitting large-scale rearrangement of the constituting helical-hairpins. This is illustrated in Fig 12 for the 100 mol% ethanol 400 ns simulation (seed 1). The initial compact packing of the helices in Fig 12A is clearly disrupted at 89 ns (Fig 12B) and at 123 ns the turn between helix IV and V is fully extended (Fig 12C). Nevertheless, N164 did not become helical, but functioned as a pivotal point connecting the two helices. Thus at 163 ns, there was a partial reconstitution of the turn, see Fig 12D.

The influence of ethanol concentration on helicity was examined using principal components analysis (PCA) applied to the helicity profiles from the simulations. However, this did not produce any clustering of similar ethanol concentrations, as demonstrated in S14 Fig in S1 File. Thus, ethanol concentration did not seem to systematically influence the distribution of helicity along the α-zein sequence. Nevertheless, the average time series plot of helicity in Fig 9D shows a slight increase (up to ∼7%) in average helicity at higher ethanol concentrations, albeit with large standard deviations. Taken together, these findings suggest a minor influence of ethanol concentration on the secondary structure in the simulations. This aligns with previous reports of constant α-helicity in the Z19 zein from maize BR451 across ethanol concentrations ranging from 28% to 70% [38]. Also, a recent circular dichroism (CD) study found no notable variations in the secondary structure of FZ3121 zein with ethanol concentrations from 60% to 90% [53].

## β-sheet

Most experimental studies on zeins have reported some β-sheet content, even for highly purified Z19 zein [36]. In contrast, most published models of α-zein monomers lack β-structure.

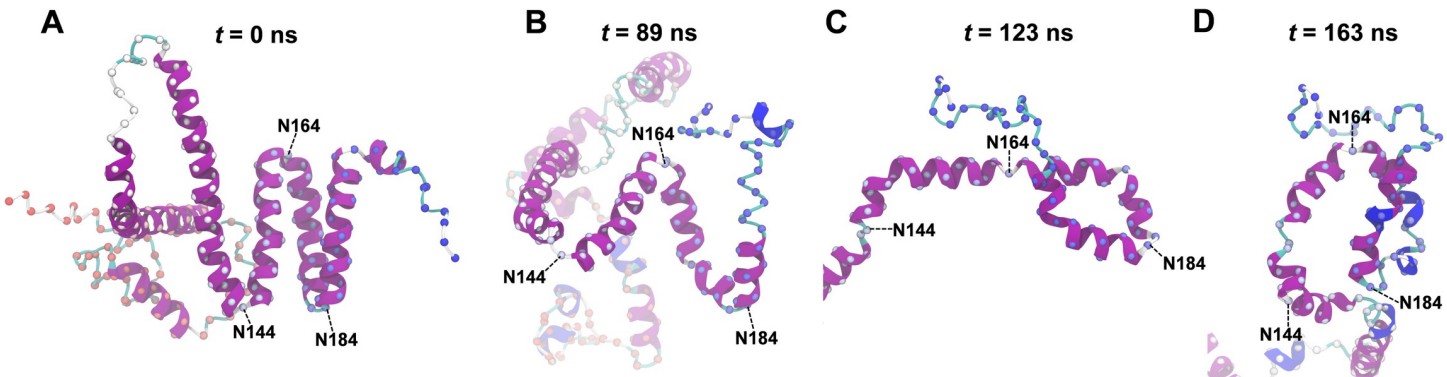

**Fig 12. Large-scale motion of helices in C-terminal bundle during a 400 ns ethanol simulation (seed 1).** A: Initial AlphaFold2 structure. B: Disruption of the α-helical bundle. C: Full extension of the helix-N164-helix hairpin. D: Partial reconstitution of the hairpin. The ribbon representation is colored by secondary structure (see legend in Fig 3B).

Mechanisms for α-helix to β-sheet transitions have been proposed for increases in solvent hydrophilicity [11] and mechanical treatment [83]. However, the extent to which β-sheet structure is inherently present in isolated α-zein monomers remains uncertain. MD simulations on isolated zeins, such as in this study, may potentially offer insights. Hence an examination of β-structure in the simulations follows below, which should be viewed in light of the potential limitations of the approach such as limited sampling and force field bias.

In all 400 ns GROMACS simulations β-sheet structure in any of the forms assigned by STRIDE (i.e., "Extended", or "Isolated bridge") was sparse and transient. This can be appreciated by considering the average content during the last 100 ns of each simulation (S4-S8 Figs in S1 File) For the water simulations, all seeds lacked extended structure while seed 1 had the highest isolated bridge content of ∼ 0.9%. For all seeds in the 2 mol% ethanol simulation series, extended structure lacked, whereas a maximum of 0.2% isolated bridge structure occurred for seed 1. Similar trends were found for the 23 mol% ethanol simulations, where extended structure was absent, but one simulation (seed 1) had 0.9% isolated bridge. The 50 mol% series lacked extended structure and had only transient populations of isolated bridge structure. Finally, for the 100 mol% ethanol series, seed 1 and 2 had 0.2% and 0.8% extended structure and 1.3% and 0.4% isolated bridge, respectively, whereas seed 3 lacked β-structure. Snapshots of the β-structure in seed 1 and 2 are shown for both simulations in S15 Fig in S1 File. For seed 1 an interaction involves β-strands formed by residues C6 –Q8 and I46 –L48. Similarly, for seed 2 there is an interaction between β-strands formed by residues G215 - G216 and Q211 - H212. These β-structures occur near the N- and C-termini, respectively, within sequence regions that were assigned coil or loop secondary structure in the AlphaFold2 model. While the 400 ns time frame apparently allows for the transition to β-structure from these relatively unstructured regions, transitioning from an α-helical structure might require longer simulations. Thus, given the potentially long timescale for transition to β-sheet structure from the initial α-helix dominant state of the AlphaFold2 model it was of particular interest to examine the last 100 ns of the three extended MD trajectories. The extension of the 100 mol% water simulation (seed 3) to 1 μs did not introduce any additional β-structure, see S21 Fig in S1 File, top panel. In case of the 2 mol% ethanol simulation (seed 2) the extension to 2 μs caused the extended β-sheet content to increase from 0.0% to 0.1% while the isolated bridge content increased from 0.0 to 1.3% (S21 Fig in S1 File, middle panel). This β-structure is described later in the section on extended simulations. For the 23 mol% ethanol simulation (seed 2) the lack of extended structure persisted until 2 μs, while there was a marginal increase in isolated bridge content relative to the first 400 ns (from 0.0 to 0.3%), see S21 Fig in S1 File, bottom panel. The small set of additional simulations carried out with the newer ff99SB*-ILDN force field [65] also showed little β-structure content, see S29 Fig in S1 File.

A brief examination of the possible induction of β-structure due to zein-zein interactions was provided by the all-atom MD simulation of 6 interacting copies of identical α-zein monomers. Although these monomers associated early in the simulation, no significant β-structure developed during the 500 ns trajectory, as evident from the time series plots in S35 Fig in S1 File. None of the monomers contained extended structure, while small amounts (up to 0.7%) of isolated bridge structures were observed for some monomers.

Thus, the MD simulations herein generally contain less than 1 percent β-structure. Several possible factors could account for this scarcity. Methodological factors might include force field biases or insufficient sampling, as the transition to β-structure may occur on longer timescales. Furthermore, specific perturbations in e.g., temperature or structure, which have not been taken into account, might also be important. On a more speculative note, one could consider that if α-zein could be isolated to the same extent as in the simulations, experimental findings might indicate a similarly low presence of β-structure.

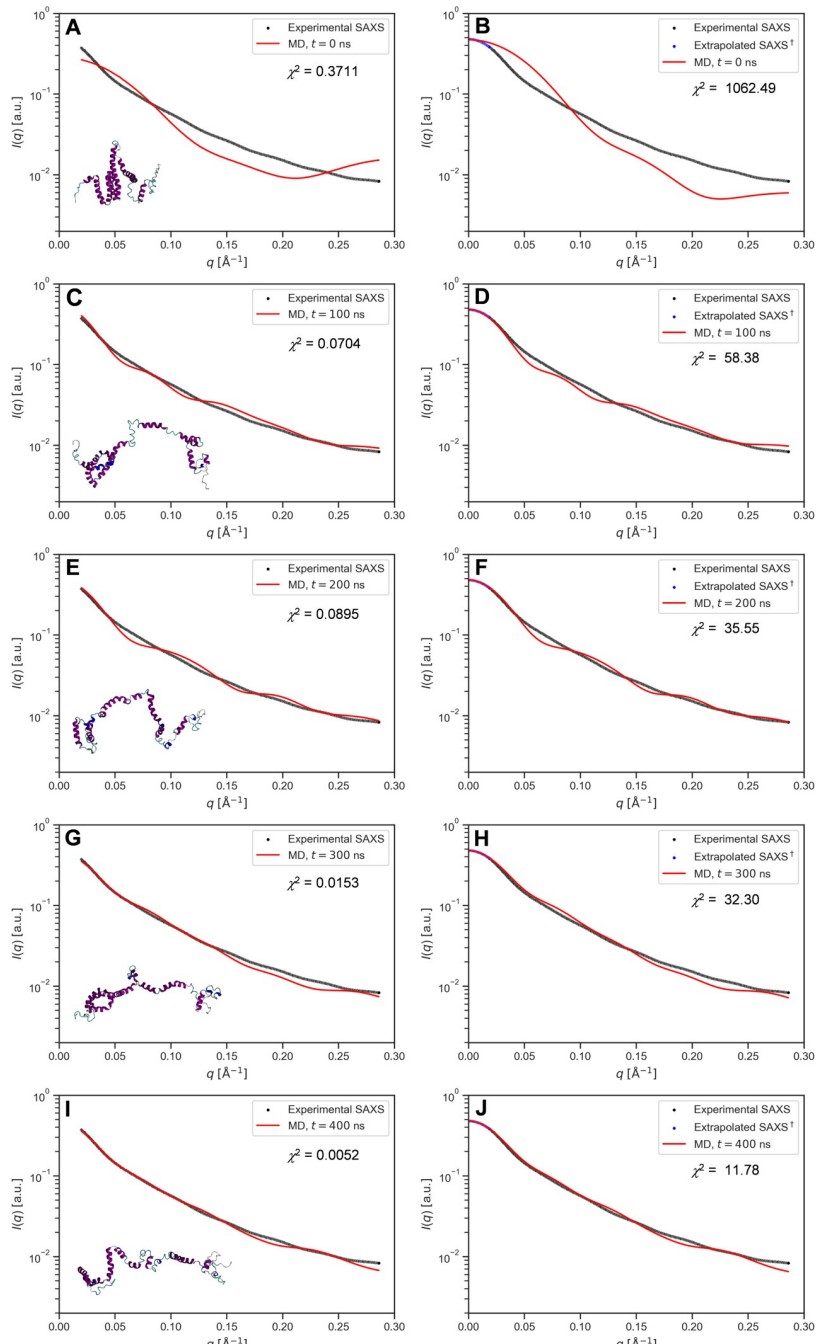

**Fig 13. SAXS predictions from MD simulations and fitting to published experimental data.** Predictions from MD frames at $t = 0$ ns, 100 ns, 200 ns, 300 ns, 400 ns of the (seed 1) 400 ns 100% ethanol simulation (red) and fit to experimental SAXS (black) from [41]. (A, C, E, G, I): MD predictions against experimental SAXS. (B, D, F, H, J): MD predictions against experimental SAXS extrapolated to $q = 0$ (blue) from the linear low $q$ region of the Guinier plot using SAXSMoW2 [84].

## Comparison with previous SAXS experiments

Small-angle X-ray scattering (SAXS) studies have provided several well-known models of Z19 and/or Z22 zein in solution. Tatham *et al.* [42] studied mixed Z19/Z22 zein from commercial

Sigma zein in 70% (v/v) methanol/water at a concentration of 8 mg/mL and modelled the solution structure as a 150 Å × 14 Å rigid rod comprised by a single α-helix without consensus-repeat segmentation. Matsushima *et al.* [43] studied a 2:1 mixture of Z19 and Z22 zeins extracted from Royal Dent 120 maize at 2 to 40 mg/mL zein concentrations in 70% (v/v) ethanol/water with added 2-mercaptoethanol to disrupt disulfide-stabilized dimers. They found that SAXS agreed with a tetramer of their ribbon like-model with dimensions 130 Å × 42 Å × 30 Å. Forato *et al.* [41] studied Z19 zein from the Z22-supressed BR451 maize strain in 90% ethanol and 10% water at a zein concentration of 3.6 mg/mL. They modelled the SAXS data as an extended 130 Å × 12 Å structure resembling a folded hairpin with mixed helix, turn and sheet content. Despite significant variation among these SAXS structures, the studies concur that α-zein assumes extended conformations in water/alcohol mixtures with a high alcoholic content.

Since only Forato *et al.* studied an isolated Z19 fraction, that work was compared with the simulations herein. The SAXS data was provided by the authors and used both as received and after extrapolation of the low $q$ range to $q = 0$ using SAXSMoW 2.0 [84] for comparison with SAXS predictions from MD simulations using FoXS [85]. Given that the authors had proposed an extended model, here an MD trajectory with notably extended conformations (100 mol% ethanol, 400 ns, seed 1) was first used for comparison. As shown in Fig 13A, the initial Alpha-Fold2 structure gave a poor fit to the experimental SAXS data. However, as the MD simulation progressed, leading to an extension of the protein, the fit improved as shown in Fig 13A, 13C, 13E, 13G and 13I. The same behaviour was observed in the fits against experimental SAXS extrapolated to $q = 0$, shown in Fig 13B, 13D, 13F, 13H and 13J. Moreover, extended snapshots from the 50 mol% ethanol simulation (seed 2) gave fits of similar good quality, see S18 Fig in S1 File. This is relevant, as the 50 mol% ethanol concentration corresponds to 78% (v/v) and thus approaches the experimental 90% concentration. In contrast, SAXS predicted for compact structures from water-rich simulations deviated markedly from the experimental curves and gave poor fits, see S17 Fig in S1 File.

## Extended All-atom MD simulations

Extended α-zein MD trajectories were analyzed in their totality, i.e., including the first 400 ns already discussed above, to identify and present key structural changes along the full simulation trajectory. To this end, MD frames were superimposed on less dynamic substructures within each trajectory. They were the α-helical bundle (N144 - D204) in case of the water 1 μs (seed 3) simulation and the 2 mol% ethanol 2 μs (seed 2) simulation. For the 23 mol% ethanol 2 μs (seed 2) simulation, helix II and III were used for superimposition.

In the 1 μs water simulation, both long helices II and III underwent almost simultaneous destabilization early in the simulation. They bent near the middle positions L74 and L124, which coincided with a decrease in α-helicity for residues L74 - L124. These residues encompass approximately the upper portion of the AlphaFold2 helix II-loop-helix III motif, with the model's orientation at 0 ns in Fig 14A. The grey arrow at 0 ns indicates the initial movement of these residues, which was part of a progression of structural condensation and association with the C-terminal α-helical bundle (N144 - D204). At 13 ns there was increased association among the C-terminal helices IV, V, VI, and VII. Furthermore, reduced distance between helix IV of this bundle and the lower helical segment of the adjacent helix III (Q133 - L143) resulted in their partial association. Also, the residues of helix II (P57 - Q97) had moved closer to the growing C-terminal helix assembly. This segment maintained considerable α-helicity and now exhibited a notable curvature, partly due to a kink involving residues L74, Q75, Q76. The curved nature of this segment gave the visual impression that it loosely encircled the

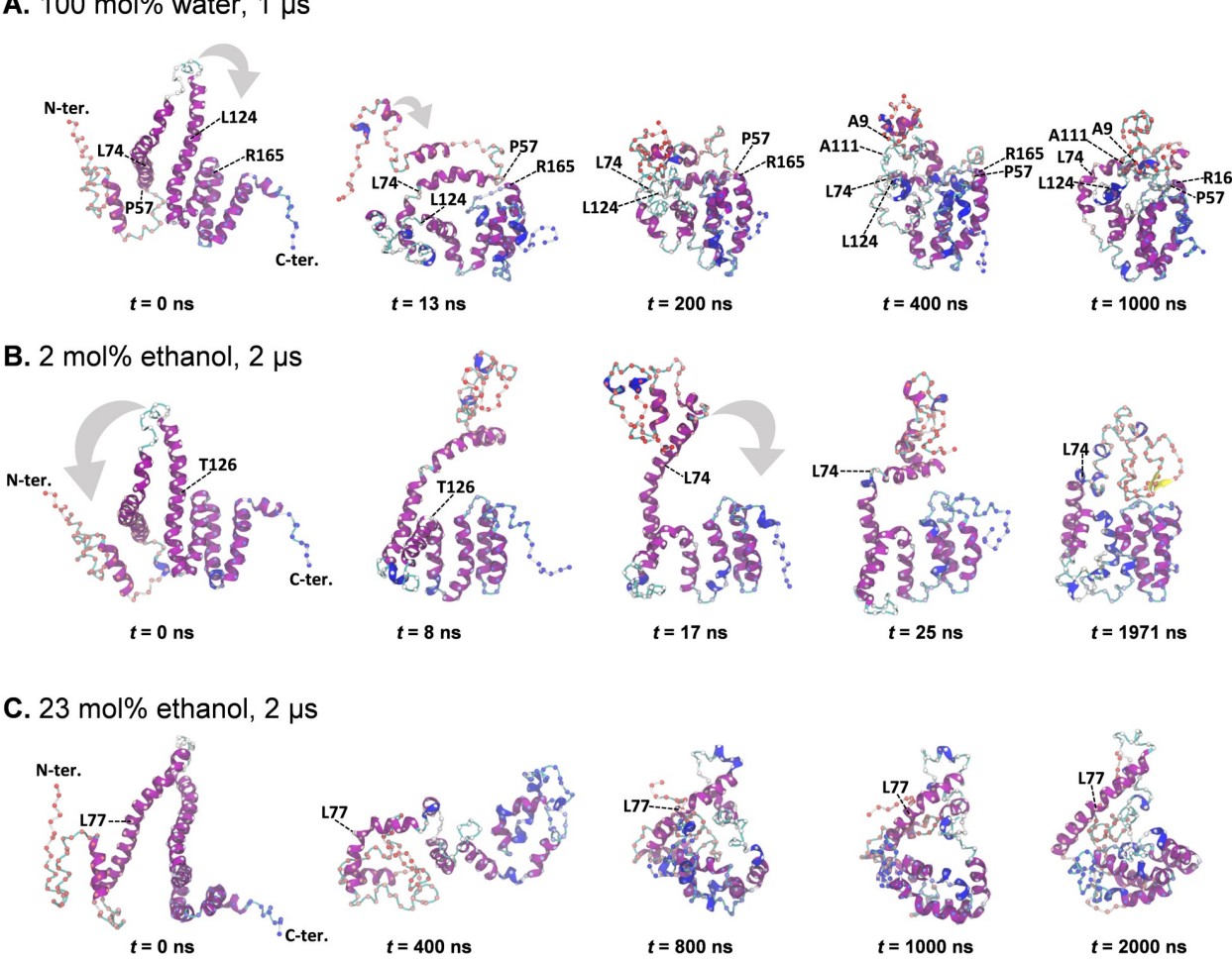

**Fig 14. Snapshots from extended MD simulations.** Grey arrows indicate approximate directions of rearrangement. Cα atoms are shown as spheres, coloured by residue position (red: N-terminal, blue: C-terminal). Labelled residues are discussed in the main text. Rotation has been used to highlight features in select snapshots, including the identical initial structures. In the case of 2 mol% ethanol (B), instead of the 2 µs endpoint the structure at 1971 ns is presented to provide a clearer view of the β-structure.

upper portion of the C-terminal helical assembly, encompassing the lower half of helix III, as well as helices IV, V, VI, and VII, extending approximately halfway around them. Additionally, the first residue of helix II (P57) had approached the first residue of helix V (R165), establishing interactions between these regions enduring throughout the remainder of the simulation. The "stickiness" of this interaction is characteristic of the type of intramolecular interactions observed in the study. From 13 ns to 1000 ns, the overall topology of the structural elements described thus far remained mostly unchanged. However, there was a continued process of condensation yielding denser packing within the lower region of the protein, as evident in Fig 14A. Alongside this condensation, notable rearrangements occurred in other regions of the structure. Particularly, the C-terminal segment (T1 - Q55) and the first residues of helix II (P110, A111) made contact at the top of the C-terminal bundle. This interaction is evident from the proximity of A9 and A111 in Fig 14A at 400 ns, and it persisted throughout the simulation. Further reorganization in the less structured upper part of the structure then took place, as reflected by the RMSD increase at around ∼500 ns (S20A Fig in S1 File) The rearrangement involved condensation, as evident from comparing the upper part of the structure

in the snapshots at 400 ns and 1000 ns in Fig 14A. This is also seen from reductions in radius of gyration and solvent-accessible surface area (SASA) in S20B, S20C Fig in S1 File. The simulation endpoint at 1000 ns had a globular conformation, characterized by a lower region containing mostly helices and a smaller and less compact upper portion with mostly coil/turn elements. Helicity was maintained in residues Q63-N83 of helix II, except for Q73 which acted as a hinge point.

In the 2 μs 2 mol% ethanol simulation, the structure rapidly converged to a compact conformation which was largely conserved for the remainder of the simulation. Thus, at 25 ns, the overall structure of the simulation endpoint at 2 μs was already mostly formed, as illustrated by the snapshots in Fig 14B. The state at 25 ns was reached by successive bending of each of the long helices III and II at hinge points roughly at the middle of each helix. First, helix III bent near residue T126, causing this helix to fold back on itself, forming a helix-turn-helix motif. This movement is indicated by a grey arrow at 0 ns in Fig 14B. The movement brought the upper half of the helix II-coil-helix III helix hairpin into lateral contact with the C-terminal helical bundle (helix IV - VI) yielding the state shown at 8 ns. This was followed by bending of helix II around L74, as indicated by a grey arrow at 17 ns. This motion placed the substructure containing residues T1 - L74 on top of the previously compacted substructure containing the remaining residues Q75 - F219. The resulting configuration at 25 ns thus appears to divide the protein into an upper part (T1 - L74) and a lower part (Q75 - F219). The most notable change in the remainder of the simulation was reorganization of the upper part, as noticed by comparing the snapshots at 25 ns and 1971 ns. The reorganization can loosely be described as a continued condensation within T1 - L74, along with their packing against residues in the lower domain. This was accompanied by a notable loss of α-helicity (from ∼70% to ∼30%) over the first 1000 ns of the simulation, followed by more stable α-helicity until the endpoint at 2000 ns, see S21 Fig in S1 File, middle panel. The loss of α-helicity occurred mainly in the upper part of the protein. During the upper part reorganization, a small amount of β-sheet was formed by the backbone hydrogen bond pairs A9/I26 and I11/A24, as shown in the snapshot at 1971 ns and in more detail in S25 Fig in S1 File.

In the 2 μs 23 mol% ethanol simulation, in line with overall simulation trends at notable ethanol concentrations, the α-zein rapidly explored of a wide range of conformations, especially in the early stages. This is evident from the pronounced RMSD fluctuations during the first ∼500 ns, see S20G Fig in S1 File. As a result, any two sparsely sampled MD snapshots were connected by a complex sequence of rearrangements that escape concise summary. However, the net rearrangement from 0 ns to 400 ns involved conversion of the middle residues (L119 - Q130) of helix III to turn, while helix II largely maintained helicity. An ensuing buckling collapse of the helix II-coil-helix III hairpin caused a downward motion of its upper part. Consequently, at 400 ns the apex residues (P110 –P120) of the hairpin appeared lodged between the N-terminal residues (T1 - N89) to the left and the C-terminal residues (S122 - F219) to the right, see Fig 14C. This unstable structure reconfigured substantially into the form at 800 ns, which maintained its overall shape until the simulation endpoint at 2 μs. The endpoint had contiguous α-helicity for residues L61-S91, thus preserving most α-helicity of the long helix II of the AlphaFold2 starting point. In contrast, helix III lost most of its α-helicity when Y113 - Q130 changed to coil. This enabled a denser packing of the involved residues, as reflected in the overall compact structure at 2 μs shown in Fig 14C. An interesting aspect of this simulation is its apparent convergence to a relatively compact structure despite significant ethanol concentration (corresponding to ∼50 vol% ethanol).

In summary, the extended simulations corroborated certain trends observed in the 400 ns simulations, notably the models' tendency to condense in water and expand with increasing ethanol concentration, along with the persistence of specific intramolecular interactions once

they were formed. This was evident in both the water and the 2 mol% ethanol simulations from the similarity between their 400 ns simulation endpoints and the structures at the end of the extended simulations, as seen in S2 Fig in S1 File. Although these persistent interactions may reflect the aggregation propensity of actual α-zeins, they pose a barrier to conformational sampling. Consequently, enhanced MD sampling studies would be an obvious extension of the present work.

Notwithstanding major structural differences, both the water and 2 mol% extended simulations gave structures with a more condensed lower region and a less dense upper region. The more compact lower region included the C-terminal α-helical bundle onto which other parts of the protein associated during the simulations. This helical bundle was relatively conserved from the AlphaFold2 model. Structural stability of the α-helical bundle in a more isolated state was further demonstrated in a 1 μs MD simulation in water on the C-terminal region N144-F219, see S6 Appendix. Additionally, as discussed above, this part of the structure mapped well onto the sequence segmentation resulting from manual alignment of cZ19C2 with the consensus sequence by Geraghty *et al.* [25].

The unusually long helices II and III in the AlphaFold2 model rearranged into shorter helical segments connected by turns or loops during the extended simulations. This was most pronounced for helix III, where no significant common stretch of residues remained helical across the three simulations. In helix II, about half of the residues maintained helicity during MD. Thus, at the simulation endpoints, predominant α-helicity was observed for residues Q63 – N83 in the water simulation, Q72 – S91 in the 2 mol% ethanol simulation, and L61 – S91 in the 23 mol% ethanol simulation.

Protein quality assessment by ProSA-web [86] gave z-scores for the extended simulation endpoints that were comparable with the experimental reference structures of ProSA, with the water simulation giving the best agreement, see S26 Fig in S1 File.

## Interactions among multiple α-zein monomers

Given that the properties of zein materials depend on interactions between zein monomers, it is of interest to characterize these interactions. While a comprehensive analysis surpasses the scope of this study, the following represents a first step towards examining interactions between the cZ19C2 model monomers. The interactions were probed by both atomistic and coarse-grained simulations of multiple copies of a globular α-zein monomer structure representative of simulations in water. As described in the Methods section, this monomer structure was obtained by subjecting the 1 μs water simulation endpoint to further 500 ns NPT equilibration in Desmond. The additional equilibration caused increased α-helical content while maintaining a globular structure overall similar to the GROMACS structure, see Fig 15A and 15B.

Interactions between multiple identical α-zeins were first studied with all-atom simulations. During MD the initial configuration (Fig 16A) rapidly formed an aggregate (Fig 16B and 16C) which remained relatively static for the remainder of the simulation. The fast aggregation was a product of the close initial proximity of the α-zein monomers and the presence of hydrophobic and polar groups on their surfaces, leading to amphiphilic association among neighboring monomers. Aggregation began as two sub-aggregates formed through the condensation of adjacent monomers. The first sub-aggregate consisted of monomers 2, 3, 4, and 5, while the second sub-aggregate contained monomers 1 and 6. The latter sub-aggregate crossed the periodic boundaries of the simulation box, leading to its interaction and association with monomers 2, 3, 4, and 5. These interactions between α-zein monomers demonstrated a persistence or "stickiness" reminiscent of the intrachain interactions noted above in monomer simulations. Thus, once again, as the interaction geometry practically becomes locked upon the first

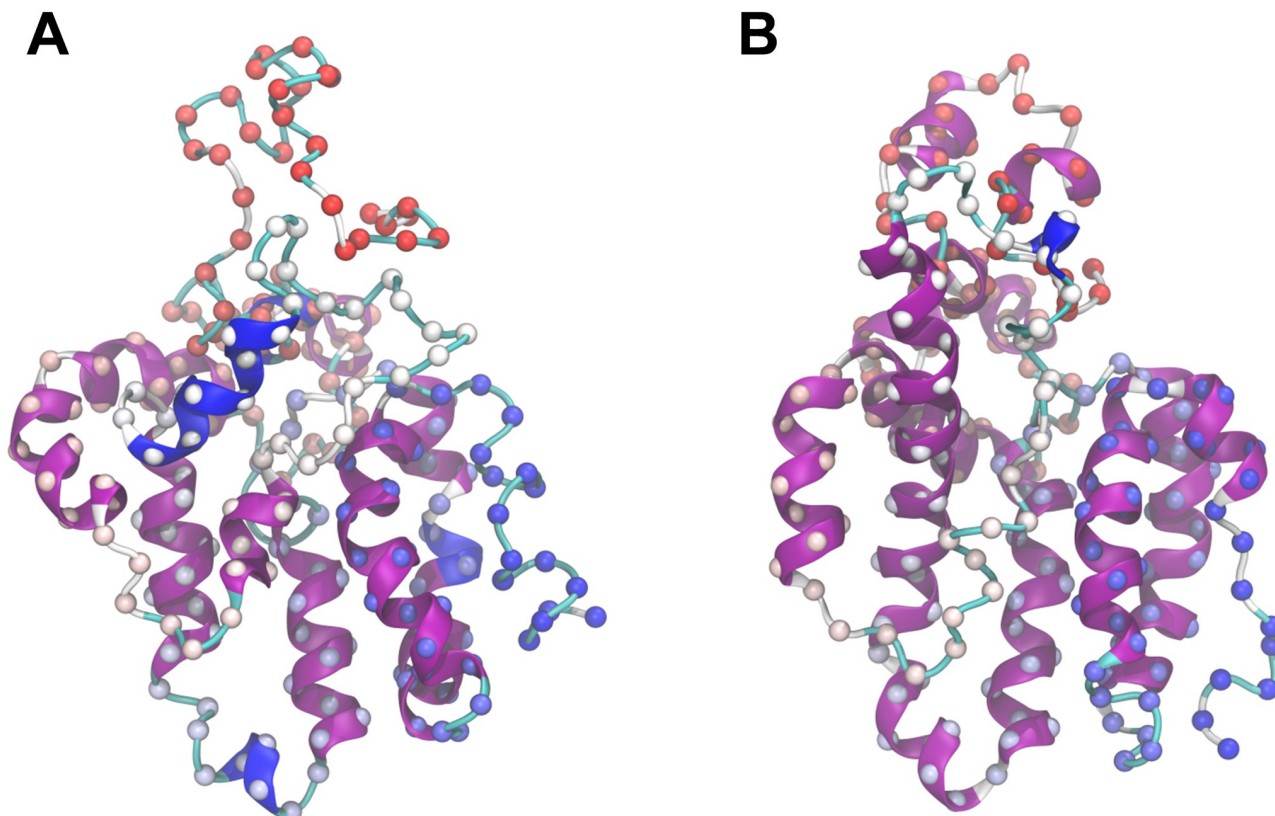

**Fig 15. Monomer preparation for multiple α-zein simulations.** A: Endpoint for the extended 1 μs GROMACS MD simulation (seed 3) in water. B: Structure after additional 500 ns Desmond MD simulation in water. This structure was used for both all-atom and coarse-grained simulations of multiple monomers. α-carbons are shown as spheres coloured by sequence position (red: N-terminal, blue: C-terminal).

monomer-monomer contact, future studies charting the landscape of zein-zein interactions will benefit from enhanced sampling techniques.

The detailed view of the simulation endpoint in Fig 16D illustrates typical hydrophobic and polar interactions between monomers in the aggregate. The interactions between monomers 1 and 6 include a salt-bridge between the side chains of 6:D204 and 1:R165, hydrophobic interactions between 6:A206, 1:L169, and 1:A166, a hydrogen bond between the sidechain of 6: N210 and the backbone oxygen of 1:L169, and additional hydrophobic interactions between residues from both monomers. The backbone RMSD timelines (S35 Fig in S1 File, second column) indicate some rearrangement within the monomers of the aggregate. Nonetheless, this was not associated with β-sheet formation, as evident from the lack of extended structure in the secondary structure timelines (S35 Fig in S1 File, first column). On the other hand, the simulated aggregate showed conserved or even slightly increased helicity in the monomers.

Oligomerization of the α-zein monomer on a larger scale was probed with coarse-grained MD simulations started from the random arrangement of 73 monomer copies shown in Fig 17C. The final simulation snapshots at 375 ns for three simulation replicates in Fig 17D–17I show non-covalent zein aggregates with both branching and linear features, resembling fractal-like aggregates [87]. The aggregates appear consistent with the known capacity of α-zeins to form viscoelastic networks in zein dough systems [88], even though the simulated structures lack β-sheets, which are presumed to have a central role in dough formation [89]. A notable limitation of the coarse-grained simulations is that the initial secondary structure assignment

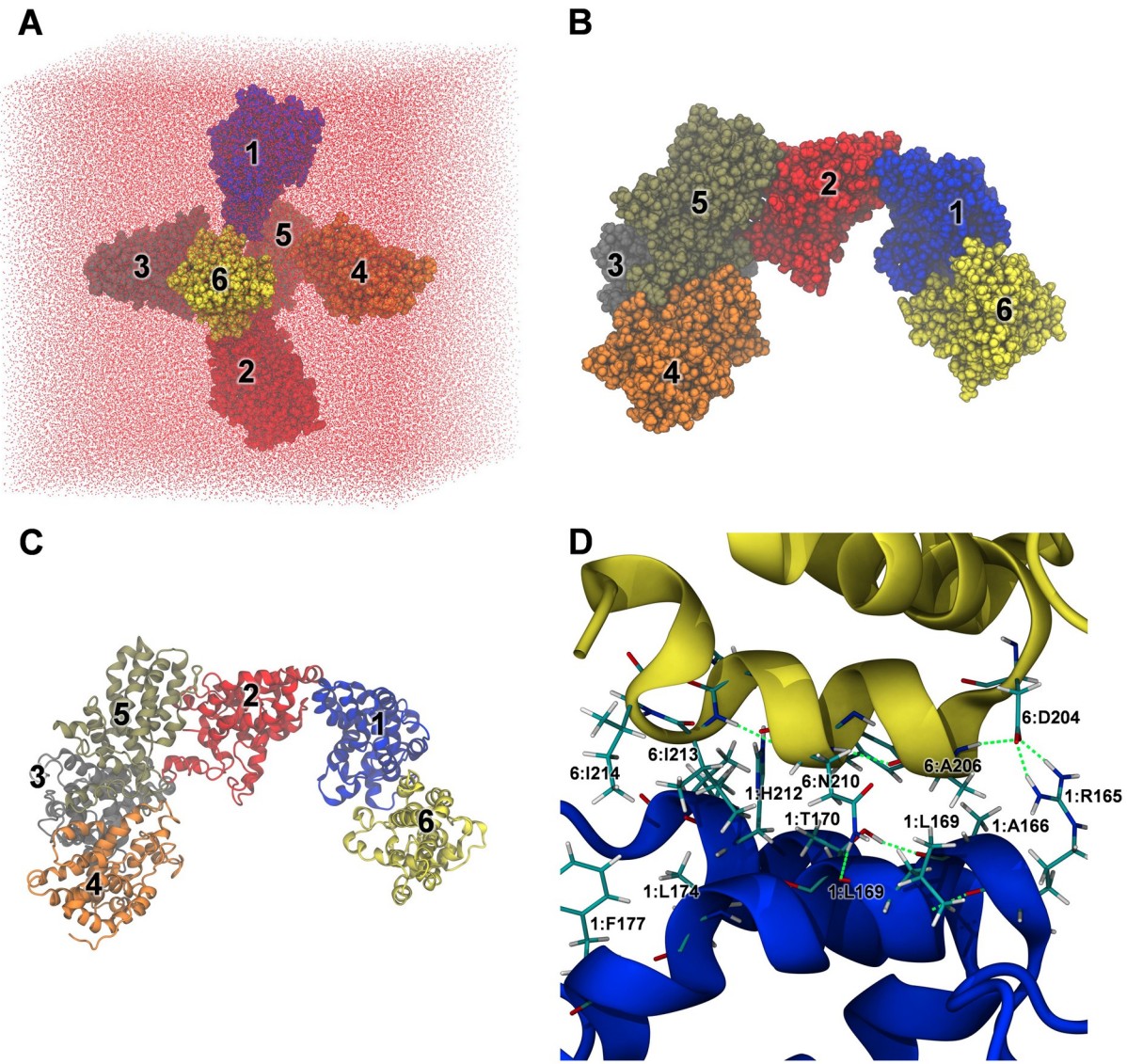

**Fig 16. All-atom MD simulations of 6 copies of the globular α-zein monomer.** A: Initial configuration. B: Structure after 500 ns MD simulation in water (space filling representation). C: Ribbon representation of (B). D: Representative interactions between zein monomers 1 and 6.

remains fixed throughout the simulations. Thus, further research is needed to explore if there is a connection between these simulated aggregates and actual zein networks.

## Potential for α-zein crosslinking

Crosslinking is widely used in protein biomaterials engineering [90] but often targets primary amines, sulfhydryl, or carboxyl groups which are scarce in α-zeins. However, the overview of α-zein amino acid composition in Table A in S1 Appendix in S1 File suggests other groups that may be modified. For instance, α-zeins contain between 7 and 10 Tyr residues that could undergo oxidative dityrosine formation [91, 92]. Additionally, 7 to 17 Ser residues could be functionalized to enable crosslinking. Moreover, although most Z19 zeins lack methionine, Z22 zeins contain 3 to 5 Met residues that might undergo oxidative crosslinking. Beyond these

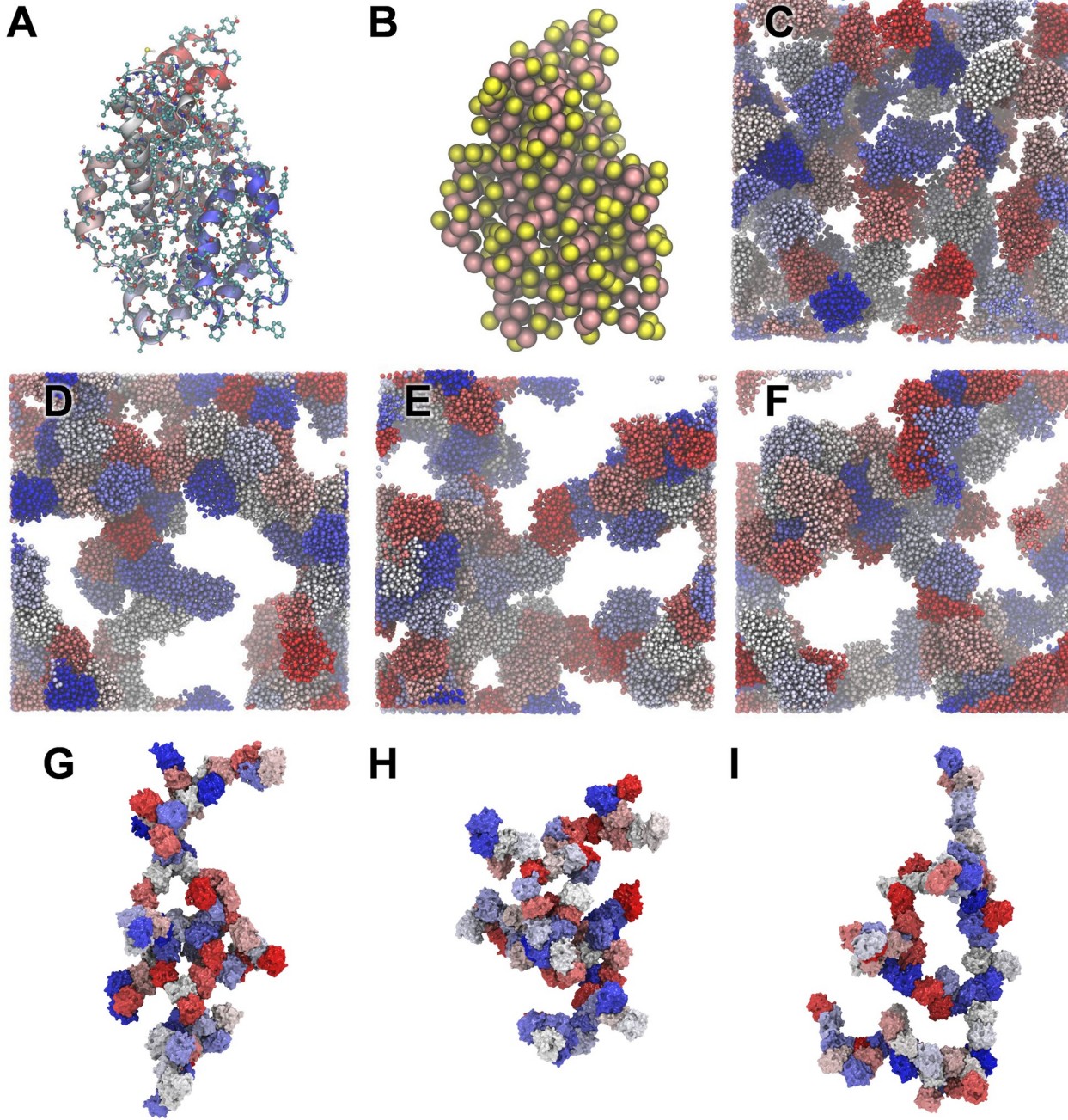

**Fig 17. Coarse grained MD simulations.** A: All-atom MD equilibrated structure before coarse graining with heavy atoms and polar hydrogens shown in ball and stick representation. B: Martini coarse-grained structure. C: Initial setup with 73 zein copies in a water box. D-F: Final MD snapshots at 375 ns for simulations started from seed 1, 2, and 3 respectively. G-I: Alternative visualization of the final MD snapshots for seed 1,2, and 3 showing the largest zein cluster across the periodic images of the simulation box. Water and chloride ions are not shown. Each zein monomer has randomly been assigned colors from a blue-red palette.

broad considerations, the current study highlighted some specific aspects regarding the cross-linking potential of the cZ19C2 α-zein via simulations, as described below.

## Cysteine-cysteine distances

The 1 to 3 cysteines in α-zeins can facilitate limited crosslinking. This is experimentally supported by Matsushima *et al.* who reported species with twice the molecular weight of Z19 and Z22 monomers for Royal Dent 120 α-zein [43]. Similarly, Cabra *et al.* reported 2–3 monomer aggregates of their purified Z19 zein in 70% (v/v) aqueous methanol at unspecified α-zein concentration in DLS and gel studies [36].

Herein, the only two cysteines in the mature cZ19C2 sequence, C6 and C27, were in close proximity in the AlphaFold2 model, with a S-S distance of 3.4 Å, indicating the possibility of an intrachain disulfide bridge. However, such short S-S distances were not maintained in the 400 ns simulations, see S16 Fig in S1 File. Instead, the average S-S distance increased with ethanol concentration, consistent with the overall protein expansion at higher ethanol concentrations noted above. Increasing structural fluctuations at higher ethanol concentrations are also noted from the S-S distance plots, where the most pronounced oscillations occurred at ethanol concentrations $\geq$ 50 mol%. For instance, in the 50 mol% ethanol (seed 3) simulation the S-S distance changed from 38.8 Å at 252 ns to 3.4 Å at 284 ns, thus transiently approaching the initial S-S distance. Given these large variations in S-S distances during MD an interesting future research direction is to explore the impact of an intrachain S-S bond between C6 and C27 on the overall conformational dynamics of the α-zein.

## Interactions with a crosslinking reagent

Introduction of methacryloyl groups is a popular route to expand the cross-linking repertoire of biopolymers while retaining biocompatibility. Additionally, this modification facilitates photopolymerization, which allows light-based 3D printing [93]. Methacrylation has been achieved for α-zein by reaction with glycidyl methacrylate (GMA) which introduced an average of 3.5 methacryloyl groups per zein monomer in mixtures of 70% ethanol and 30% water [94]. The authors noted lower methacrylation at higher ethanol concentrations. They attributed this reduction to a decrease in solvent polarity, rather than a reduced exposure of reactive groups. This interpretation finds support in both the current and prior studies, as they show that higher ethanol concentrations cause an increase rather than decrease in overall structural exposure. Conversely, since simulations in solvent with high water content gave compact α-zein conformations, it became intriguing to investigate the compatibility of these conformations with the experimentally demonstrated GMA functionalization in hydrophilic environments.

MD simulations on the compact α-zein model in water containing GMA demonstrated both exposure of α-zein polar groups and their interactions with GMA. This is illustrated in Fig 18 by the time development of the number of GMA molecules within 4 Å of the protein (green) or beyond (red). The top left inset figure shows the initial spherical placement of 48 GMA molecules around the α-zein and the top right inset figure shows the simulation endpoint where multiple GMA molecules are associated with the protein. The last 100 ns of the time series showed an average partitioning of 26 GMA molecules (55%) within 4 Å of the protein and 22 (45%) beyond. Representative snapshots of interactions of long duration between GMA and α-zein are shown in Fig 19. In the first case (Fig 19A) several hydrophobic/amphiphilic interactions with P30, A112, L119, and other residues tether the GMA molecule to the zein surface site for $\sim$50% of the simulation time. The hydroxyl oxygen of S122 was within 2.9 Å of an GMA epoxide carbon. This configuration may potentially lead to epoxide ring opening and subsequent functionalization. Another notable interaction between α-zein and GMA is shown in Fig 19B. Here, GMA was associated for $\sim$15% of the simulation time with the hydrophobic patch formed by A71, L74, L77, V78, L81, A125. This placed the hydroxyl oxygen

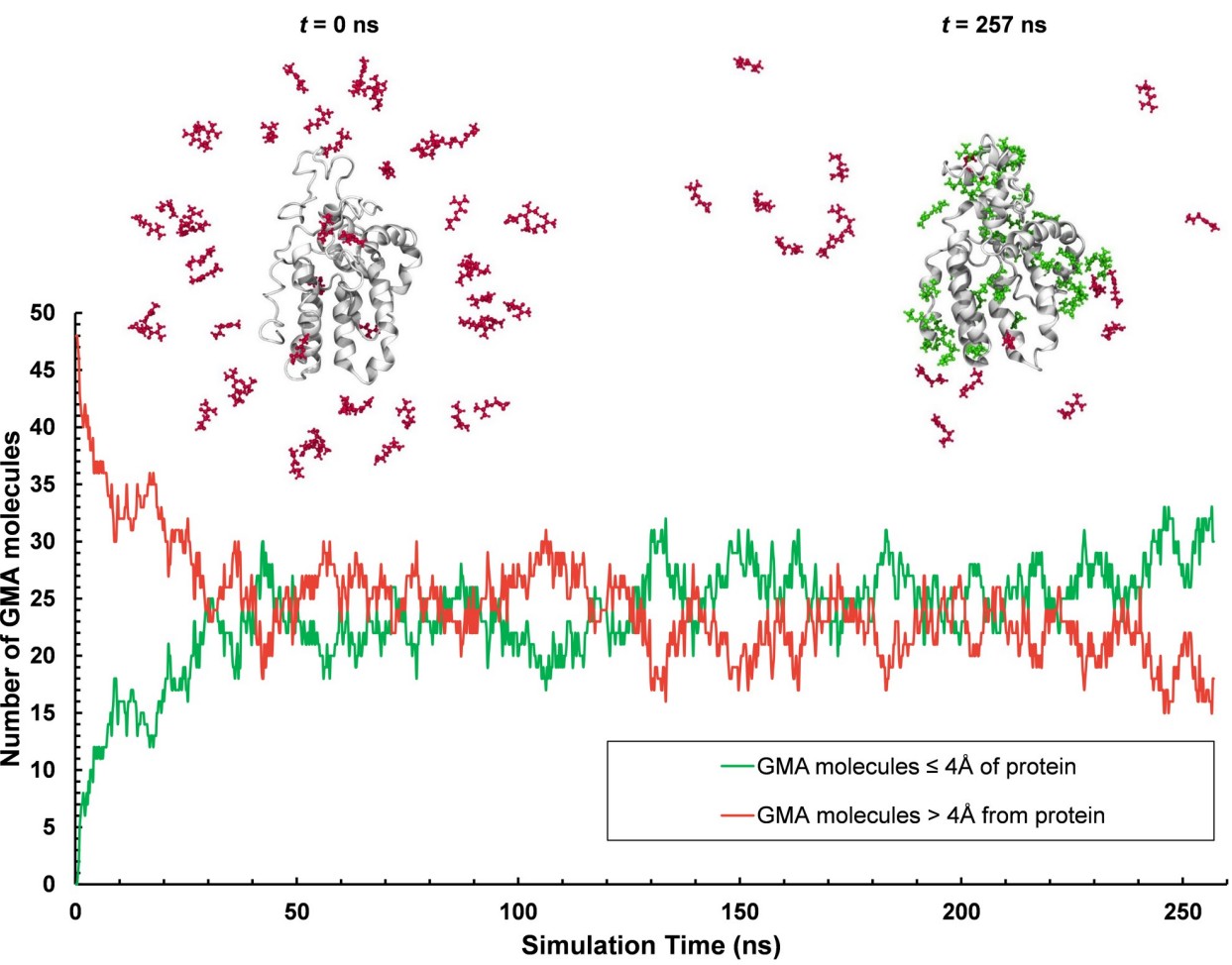

**Fig 18. Interaction between glycidyl methacrylate (GMA) and α-zein during MD.** The inset structures show the initial ($t = 0$ ns) and final ($t = 257$ ns) configuration of α-zein (white ribbons) and 48 GMA molecules. Red indicates GMA molecules more than 4 Å from the protein. Green indicates GMA molecules within 4 Å of the protein.

of Y128 within 3.3 Å of a GMA epoxide carbon, which may again lead to a pre-reaction complex for functionalization.

## Conclusions

The α-zein cZ19C2 which is one of the three most highly expressed clones previously identified [30] was modelled with AlphaFold2. This gave a structure with 7 loosely packed α-helical segments connected by turns or loops, with a total of 68% α-helicity and no β-structure. Compact tertiary structure was limited to a C-terminal bundle of three α-helices. Notably, there was a strong correspondence between the individual α-helices in this bundle and the consensus repeat sequence NPAAYLQQQQLLPFNQLA(V/A)(L/A) by Geraghty *et al.* [25]. This marks a deviation from previously published models [28, 35, 43], based on the consensus sequence proposed by Argos *et al.* [28].

To systematically examine the impact of solution environment, the AlphaFold2 model was used as a starting point for multiple 400 ns MD simulations. Triplicate simulations (seed 1, 2, and 3) differing by random initialization of atomic velocities were conducted in a mixed water/ethanol solvent, with varying concentrations of ethanol (0 mol%, 3 mol%, 23 mol%,

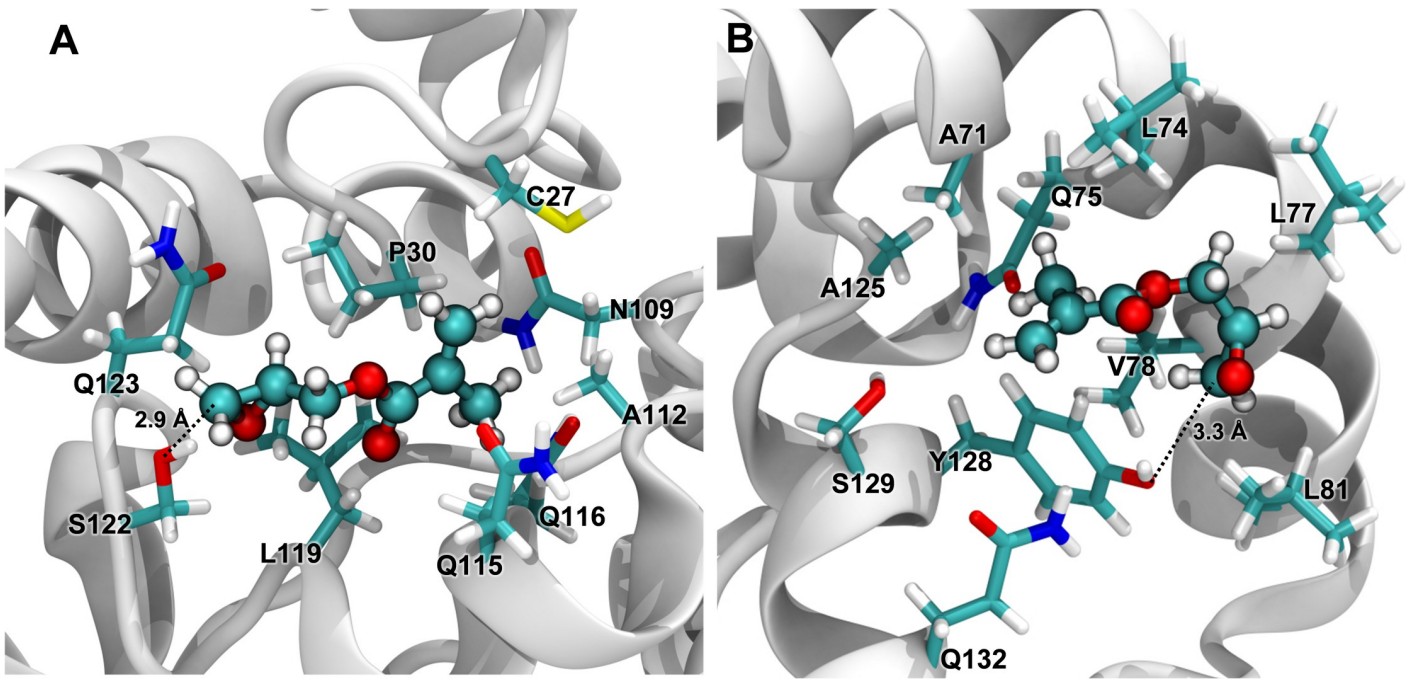

**Fig 19. α-zein sites with high residence times for glycidyl methacrylate (GMA) during MD.** GMA is shown as ball and stick models, zein residues within 4 Å of GMA are shown in stick representation, zein backbone shown in ribbon representation. **A:** GMA binding persisting for ∼ 50% of the simulation time. **B:** GMA binding persisting for ∼ 15% of the simulation time.

50%, and 100 mol%). Selected simulations (seed 3 for 100 mol% water, seed 2 for 2 mol% ethanol, and seed 2 for 23 mol% ethanol) were extended to 1 μs, 2 μs, and 2 μs respectively, to enhance sampling. All simulations gave markedly different structural endpoints, reflecting the loosely packed initial structure's sensitivity to perturbations. In water and low (≤ 2 mol%) ethanol concentration, the α-zein model rapidly collapsed to compact structures that reorganized little and slowly during the remainder of the simulation. In contrast, higher ethanol concentration was associated with structural expansion and increased conformational sampling. Tertiary structure, in particular the C-terminal helical bundle, was partially preserved in water and at low to intermediate ethanol concentrations. At ethanol concentrations ≥ 50 mol%, tertiary structure was mostly or entirely disrupted.

Elongated structures from MD simulations with high ethanol concentrations gave the best agreement with previously published SAXS experiments. In contrast, compact conformations from simulations with high water content did not align with the experimental data. These findings corroborate the previous hypothesis [41] that α-zein predominantly assumes elongated and flexible conformations in a binary ethanol/water solvent with a substantial ethanol concentration.

Principal components analysis did not reveal a clear coupling between ethanol concentration and the percentage distribution of secondary structural elements (helix, sheet-, etc.). However, averaged time series data indicated a subtle overall helicity increase with ethanol concentration.

Interactions among identical zein monomers were briefly examined with both all-atom and coarse-grained simulations using the globular structural endpoint of the 1 μs water simulation (seed 2) after further 500 ns equilibration with the alternative force fields OPLS3 and TIP3P. In all-atom MD simulations in water, 6 copies of the monomer rapidly formed an aggregate.

Multiple amphiphilic interactions between monomers stabilized the aggregate, hindering reorganization during the remainder of the simulation. Larger scale aggregation was probed by coarse-grained MD simulations on 73 α-zein monomers in water, yielding branched non-covalent networks of monomers.

In contrast to most published experiments, very little (generally < 1%) β-sheet or isolated bridge structure was found in any of the all-atom simulations (fifteen 400 ns OPLS-AA/SPCE simulations of which three simulations were extended to the microsecond timescale, four additional ff99SB*-ILDN/TIP3P simulations, and one OPLS3/TIP3P simulation of 6 interacting zein monomers in water.) Although this lack agrees with most published α-zein models, more exhaustive conformational sampling is needed to ascertain if this reflects an inherent low β-sheet propensity for the studied α-zein.

Interactions between the common crosslinking agent glycidyl methacrylate (GMA) and a compact, globular α-zein conformation representative water simulations were probed by all-atom simulations in water. After equilibration, the total 48 GMA molecules initially located $\geq$ 4 Å from the protein were distributed with 26 GMA (55%) within 4 Å of the protein and 22 (45%) beyond, suggesting high affinity between GMA and compact α-zein conformations in water. Inspecting two specific GMA-zein interactions persisting for 15% and 50% of the simulation time, showed arrangements where a GMA epoxide carbon was within 3.3 Å and 2.9 Å of the hydroxyl group oxygens of Y128 and S122, respectively. These are potential pre-reaction complexes for methacrylation of α-zein.

Overall, the loosely packed AlphaFold2 model allowed relatively unhindered reorganization on the timescale of the simulations, thus providing a platform for α-zein conformational exploration. Attractive interactions established in water-rich simulations both within the zein monomer and between zein monomers tended to be highly persistent or "sticky". While possibly reflecting the known aggregation propensity of α-zeins, future studies may probe the robustness of these results and examine their connection to experiments. The simulations suggest directions for further computational studies. For instance, the use of long simulations and multiple seeds/replicates for the studied α-zein model likely extends as a guideline for α-zein models in general. In addition, "sticky" interactions may potentially be overcome by enhanced sampling methods. Future simulations may also explore alternative protonation states, and the impact of a disulfide bond between C6 and C27. Finally, as only a single highly expressed α-zein (cZ19C2) was studied, a next step towards computational elucidation of practical zein mixtures could entail simulations on the other two highly expressed zeins clones [30].

In summary, the modelling approach for the cZ19C2 α-zein monomer herein offers a new perspective relative to previous α-zein models, by subjecting an initial AlphaFold2 structure to thorough MD sampling in binary ethanol/water solvents. Instead of approximating a single "native-like" structure, this study presents ensembles of α-zein conformations characteristic for each ethanol concentration. Such ensembles, together with the body of previous α-zein models, contribute to an increasingly detailed map of α-zein conformational variation across a multitude of environments. This knowledge may e.g., inform strategies for introducing chemical modifications to the α-zein monomer, and in turn pave the way for advances in α-zein engineering, including the development of new biomaterials.

## Supporting information

**S1 File.**
(ZIP)

**S1 Striking image.**
(TIF)

## Author Contributions

**Conceptualization:** Niels Johan Christensen.

**Formal analysis:** Niels Johan Christensen.

**Funding acquisition:** Niels Johan Christensen.

**Investigation:** Niels Johan Christensen.

**Methodology:** Niels Johan Christensen.

**Writing – original draft:** Niels Johan Christensen.

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
