## [Decision Letter · Decision Letter 0]

2 Jan 2024

PONE-D-23-34203Conformations of a highly expressed Z19 α-zein studied with AlphaFold2 and MD simulationsPLOS ONE

Dear Dr. Christensen,

Thank you for submitting your manuscript to PLOS ONE. After careful consideration, we feel that it has merit but does not fully meet PLOS ONE’s publication criteria as it currently stands. Therefore, we invite you to submit a revised version of the manuscript that addresses the points raised during the review process.

As you can see below, Reviewers #1, #2 and #4 are entirely or partially positive on your work. Although Reviewer #4 recommended major revisions, the amount of work required is not dramatic and would critically improve the quality of this Manuscript. However, Reviewer #3 strongly recommended major revisions (points 1.1, 1.2 and 1.3) and in particular what brings the most perplexity and should be principally addressed is point *1.2) Second major revision: quality of the a-zein AF2 predicted 3D model.* Considering the tremendous amount of work that has been conducted, my suggestion is to do analyses of the C-ter three a-helix bundle only in addition to those already obtained on the full-length and discuss further the results in light of the general points raised out by this Reviewer on the model quality.This manuscript is of interest for this Journal and the scientific community, as Reviewer #2 pointed out, *it is a comprehensive study describing the landscape of *possible* structures for further investigation*.

We look forward to receiving your revised manuscript.

Kind regards,

Matteo De March

Academic Editor

PLOS ONE

Journal Requirements:

Reviewers' comments:

Reviewer's Responses to Questions

**Comments to the Author**

1. Is the manuscript technically sound, and do the data support the conclusions?

Reviewer #1: Yes

Reviewer #2: Yes

Reviewer #3: No

Reviewer #4: Yes

2. Has the statistical analysis been performed appropriately and rigorously? 

Reviewer #1: Yes

Reviewer #2: Yes

Reviewer #3: No

Reviewer #4: N/A

3. Have the authors made all data underlying the findings in their manuscript fully available?

Reviewer #1: Yes

Reviewer #2: Yes

Reviewer #3: Yes

Reviewer #4: Yes

4. Is the manuscript presented in an intelligible fashion and written in standard English?

Reviewer #1: Yes

Reviewer #2: Yes

Reviewer #3: Yes

Reviewer #4: Yes

5. Review Comments to the Author

**Reviewer #1**: In this work, the Author characterized the structural ensemble of alpha-zein cZ19C2 using a model predicted by AlphaFold2 (AF2), followed by all-atom and coarse-grained MD simulations on the model. Started with a loosely packed initial AF2 model, the protein exhibited high conformational heterogeneity in simulations, suggesting that ensemble-based approaches are preferred over single native-structure based ones. Valuable observations were obtained – including expanded conformations in high ethanol conditions.

Overall, this manuscript presented substantial and valuable results in many aspects. There are, however, several points that require attention:

1. The MD simulations suggest the C-terminal helical bundle has a relatively stable structure compared to the rest of the protein, Figures 14A and B may indicate a disordered N-terminal region and a more stable C-terminal core region. Also, sequence from segment 6 to C-terminal end agrees well with the consensus repeat (Figure 3).

In light of these, I wonder if AF2 prediction on the shortened sequence (either from segment 6 to C-terminus or helices 4-6) will yield a structure with better confidence. If yes, it may be interesting to model these regions separately before their integration, perhaps in future studies.

2. What is the rationale of using OPLS forcefield? Does it provide a more accurate modelling on proteins with disordered regions such as the one studied here?

3. It may be interesting to follow and elucidate conformational changes and key molecular interactions near the point of divergence in different replicates at early time points in the simulation. Besides Figure 14, a detailed comparison at early time points between replicates, in various conditions, will enhance the manuscript.

Minor points concerned with general layout and illustrations:

4. In my opinion, the Abstract contains a lot of technical quantitative descriptions, that would be better replaced with more concise descriptions on major findings and their implications, in order to attract readers.

5. There are many brief sections in Methods that should be merged. For example: “GROMACS All-atom Simulations” & “GROMACS All-atom Simulations Protocol”.

6. Typological error at line 318 “…the sequence bas been…”

7. Similar to 5, the Result part of the manuscript would benefit from merging related sections, if appropriate. In general, the readability could be better improved with a more condensed layout and descriptive section titles.

8. In my opinion, some main figures could be combined to reduce redundancy, for example Figures 4-8. There are also too many Supplementary Figures (36), this number should be reduced

9. In Figure 11, the uniform orange “bars” can be confusing, they may be better represented by other means.

10. To help visual comparison of these highly heterogenous structures in most figures, coloring the ribbon presentation of protein structures by residue numbering (such as S2 Appendix C) may further enhance the quality of these illustrations.

**Reviewer #2**: This paper leverages Alphafold2 to generate a model of a representative alpha-zein and then performs molecular dynamics simulations in modelling structures in different concentrations of solvent and makes some attempt to identify interfaces that may exist in oligomers. It correlates results of the simulations with SAXS data and reports good correlations for some endpoints. The length of the simulations is adequate to converge on structures within runs. However the endpoints are diverse. It is not clear whether the structures are reasonable or relevant to structures adopted in a physiologic setting, but the author has not claimed that this is the case. As such, it is a comprehensive study describing the landscape of *possible* structures for further investigation. Given that alpha-zeins may be useful biomaterials this information may inform production pathways. The manuscript is clearly written, the data supports the modest claims, and comparisons of the current study with past literature are outlined.

**Reviewer #3**: 2023/12/15

Conformations of a highly expressed Z19 a-zein studied with AlphaFold2 and MD simulations

Niels Joan Christensen

University of Denmark

The main goal of the manuscript is to study the conformation landscape of the cZ19C2 a-zein, i.e. a 240 aa protein which is expressed in maize. The protein shows various applications as a biopolymer. Bioinformatics sequence analysis and molecular modeling are both valuable approach to study the a-zein since the protein has been refractory to 3D structure resolution by classical experimental methods (NMR, X-ray crystallography).

General remarks:

The english of the manuscript is of good quality. The a-zein is a particularly challenging protein to study since no homologous structure is available in the protein data bank (PDB). The author did a tremendous amount of work, especially molecular dynamics (MD) simulations.

When writing a scientific article, concision is appreciated. The author presents 19 figures in the main manuscript and 36 in Supplementary information (SI). For the sake of clarity, 7 or 8 figures in the main and 15 in SI should not be exceeded otherwise the manuscript may appear bushy. The graphical quality of the structural pictures were rather poor and improvements are needed. Residues of Figure 3a were also difficult to read, for instance I had difficulties to distinguish Ile (I) and Leu (L) residues. Did the author make screen captures of pictures ? If yes, it should be avoided for publication because the output is of low graphical quality.

I was interested by the article because a-zeins are challenging bioinformatics targets and I found original that some MD simulations were carried out in ethanol rather than water as it is traditional. I was surprised that the author wrote a long conclusion whereas the manuscript has no discussion. This must be corrected. I understood that the author’s main claim was that the protein adopts a packed conformation in water and an elongated one in ethanol. However I have both questions and major concerns about the data which is presented and I wish the author will find them valuable to increase the quality of the manuscript.

1) Major revisions

1.1) First major revision: The a-zein 20 amino-acid repeat

The cZ19C2 a-zein is a member of a 100 protein coding gene family. The a-zeins present a 20 amino acid repeated sequence which seems to play a crucial role in the protein folding and functional organization. The author mentioned that there is no general agreement on the definition of the repeat unit, therefore the reader wishes that his work will clarify this point. Using a multiple alignment of eight repeats, Geragthy D et al (1981) proposed a sequence consensus while Argos P. et al (1982) published an alternative consensus using a 18 repeat sequence alignment. After I consulted both referenced articles, I noticed that Geragthy’s and Argos consensi are shifted.

So, which consensus is correct ? In the introduction the author quotes both Argos (line 84) and Geragthy’s (line 81) consensi. Unlike Geragthy, Argos is said to have been largely referenced (line 85). However, the author will use Geragthy’s consensus (Abstract line 31, Results line 319) for the present study without clear justification. As a result, the reader is puzzled and confused right from the start.

At the sequence level and because there is no homologous structure, a crucial task is to predict how many repeats are present in the cZ19C2 sequence, where they are localized and to which degree each of them is similar to the consensus. In figure 3a, the author proposed a representation that is too qualitative and to some extend outdated to estimate the similarity between the a-zein repeats and the consensus, he does not provide a quantitative measurement. For instance, segments 1-5 and 12 are not convincing repeat sequences because visually they poorly match the consensus.

As a first major revision, the author should construct both a repeat LOGO (Crooks GE et al, 2004) (https://weblogo.berkeley.edu/examples.html) which is nowadays a dramatically better representation than a consensus and a a position weight matrice (PWM) (Position_weight_matrix in https://en.wikipedia.org/wiki/). A quantitative similarity measurement between each cZ19C2 repeat and the PWM can be calculated using the matrix similarity score (mSS) that was described in the MATCH article (Kel A.E.et al, 2003, DOI:10.1093/nar/gkg585).

I expect also the author to explain why he prefers Geragthy’s consensus rather than Argos one. The position of the repeats in the AlphaFold 2 (AF2) 3D model should be also clearly represented. A crucial result is to determine if AF2 predicts a-helices for the repeats.

1.2) Second major revision: quality of the a-zein AF2 predicted 3D model

At the structural level, the author starts the conformation investigations using a 3D molecular model of the cZ19C2 protein generated by ColabFold, i.e. a web interfaced AF2 server and hypothesizes that deep learning might compensate for the lack of a-zein homologous protein experimental structures in the Protein Data Bank (PDB). I am afraid the author assumed too much. Since AF2 breakthrough at CASP14, it has been known that AF2 generates good, bad and ugly models (see J.M. Thornton et al, 2021, DOI:10.1038/s41591-021-01533-0). The cZ19C2 a-zein model belongs to the group of bad or ugly protein 3D structure predictions. I think it is reckless to embark on MD simulations and conformation studies using a 3D model that shows such a poor quality unless the author proved that the MD were long enough to simulate the 3D folding of the a-zein polypeptide primary sequence. Since the author did not claim it, I understand it was not his ambition. Two AF2 metrics warned that the model was bad, first the per-residue LDDT scores and second the predicted aligned error matrix (PAE) matrix, the latter was totally absent in the manuscript. To study the conformation landscape of a protein, it is crucial that a reliable model or experimental structure is available. For the a-zein it is definitely not the case. The per-residue LDDT scores are low or very low. Only the C-terminus three helix bundle may be valuable – see below-. First, I request the author to create a main figure that will present a cartoon of the 3D model with each residue colored according to its LDDT score. The reader must know if the AF2 model can be trusted and this information must appear graphically on a structural figure in the main manuscript. Second, the author must also show in the same figure the PAE that is generated by AF2 to determine how reliable are inter-helix distances in the model. Using the PAE of the a-zein AF2 model (see P06677 record in the EBI AlpaFold database) , the matrix is largely white, i.e. AF2 does not know how to organize spatially the secondary structures. It is unfortunate that the author did not take into account the PAE as a second bad quality warning of the model. It can be observed that the matrix is slightly greener in the 160-225 residue square, i.e. the fold of the C-terminus three helix bundle. This region is slightly more reliable than any other part of the model which is totally unusable for further analysis. As a consequence, the MD simulations and the conformation study should be carried out only on the helix bundle. This is my major remark since I consider that outside the bundle the quality of the AF2 model is so bad that running MD simulations makes no sense.

1.3) Third major revision: Write a discussion

The article shows no discussion paragraph. The author is expected to discuss his results, for instance the quality of the AF2 model.

1.4) Fourth major revision: number of figures in the main manuscript and SI

For clarity and concision, reduce to 7 or 8 the number of figures in the main manuscript while 15 is recommended not to be exceeded in SI.

2) Minor revisions

2.1) Abstract:

In the cell, what is the natural environment of a-zein, i.e. cytoplasm, membrane, extracellular ?

What is glycidyl methacrylate ? Do not assume that readers will know what is this compound for.

2.2) Expressed sequence tags (ESTs)

Unlike RNA-seq reads, ESTs are not correlated with gene expression levels, they enable the biologist to determine which genes are expressed in a tissue sample and only provide a qualitative information on which genes are highly expressed. Therefore “The top-three expressed random selected clones measured by percentage expressed sequence tags (ESTs)” sentence is not rigorous and an alternative formulation should be found such as

“Using EST, az19B1, az19B3, and az22z1 were found among abundantly expressed genes”.

2.3) Figure 1

The percentage values that are indicated on Figure 1 tree should be withdrawn since the percentages are not correlated with gene expression levels. Mark the three selected a-zein with a symbol to indicate that they are among abundantly expressed genes in B73 endosperm, it would be enough. In the tree, why do the cZ19C2 leaf shows two UniprotKB identifiers ? Is it a sequence redundance in UniprotKB ? Please select only one for clarity.

Was the a-zein tree built using a protein sequence alignment (which program ?) and a phylogenetic algorithm (NJ, Bayes, ML, MP) ? In the legend, please give more information on the method. The tree lacks a scale to understand what are the numerical values on the branches.

2.4) Figure 2:

What do numbers in brackets represent ? Please write it in the figure legend.

2.5) Figure 3A:

The residue offsets do not correspond to sequence Uniprot-ID:P06677.

I understand the author trimmed off the Nter peptide signal and renumbered the residues starting from Ala31. This is confusing when the readers wish to localize precisely the repeats in the Uniprot-ID:P06677 sequence. Renumber the residues in figure 3A using P06677 full-length protein sequence.

2.6) Figure 4 to 8:

The structural figure resolution is too low to clearly distinguish the conformations.

The helices are colored in magenta but some regions are colored in blue or yellow. The author does not explain in the legend what does the blue and yellow colors mean.

Figure 7E, 7H, 8B, 8E and 8H are particularly messy.

3) Reviewing conclusion:

Because of the tremendous amount of work which has been produced by the author I would feel guilty if I suggested the manuscript to be rejected. However, there are serious concerns about the reliability of the AF2 model which was used as the starting structure for MD simulations and major revisions are required. On one hand, claiming that the a-zein shows a packed conformation in water sounds to me as an expected consequence of the poor quality of the AF2 model. Indeed, the three MD seeds of figure 4 in 100% water ends with protein conformations that cannot be superposed even locally, i.e. helices are messily packed. Don’t we expect some kind of structural ordered organization of the protein to understand biopolymer properties ? On the other hand, to claim that the polypeptide elongates in ethanol reminds me the protein denaturing effect of the solvent. Is it an original result ? I would say no. Therefore major revisions must be carried out and I invite the author to rethink the article by focusing the analysis on the C-terminus three a-helix bundle only.

**Reviewer #4**: The authors have performed an interesting work on Conformations of a highly expressed Z19 α-zein studied with AlphaFold2

2 and MD simulations. However, a few queries need to be addressed.

1. The title seems to be less scientific. The authors can reframe the title as "Insights into Z19 α-Zein's Structural Dynamics: AlphaFold2 and MD Simulation Approaches".

2. What motivated the selection of ethanol concentrations (0, 3, 23, 50, and 100 mol%) in the MD simulations? Were these concentrations based on previous studies or theoretical considerations?

3. What is the significance of the proposed work? What is the cause behind employing various force fields in MD simulation?

4. How was the reliability or accuracy of the AlphaFold2 model assessed prior to commencing the molecular dynamics simulations?

5. Could you provide insights into why extended conformations prevailed at higher ethanol concentrations and the impact of these conformations on the protein's structure and function?

6.Beyond biomaterials, are there other industrial or environmental applications where this detailed map of α-zein conformational variation could be particularly relevant or impactful?

7. Why did the authors focus on backbone RMSD? What is the reason behind employing 400 ns simulation period?

8. Explain how did the authors declare collapse and structural expansion based on MD simulation

6. PLOS authors have the option to publish the peer review history of their article (what does this mean?). If published, this will include your full peer review and any attached files.

Reviewer #1: No

Reviewer #2: No

Reviewer #3: No

Reviewer #4: No

---

## [Author Response · Author response to Decision Letter 0]

28 Feb 2024

Response to reviewers (PONE-D-23-34203)

Dear Dr. Christensen,

Thank you for submitting your manuscript to PLOS ONE. After careful consideration, we feel that it has merit but does not fully meet PLOS ONE’s publication criteria as it currently stands. Therefore, we invite you to submit a revised version of the manuscript that addresses the points raised during the review process.

As you can see below, Reviewers #1, #2 and #4 are entirely or partially positive on your work. Although Reviewer #4 recommended major revisions, the amount of work required is not dramatic and would critically improve the quality of this Manuscript. However, Reviewer #3 strongly recommended major revisions (points 1.1, 1.2 and 1.3) and in particular what brings the most perplexity and should be principally addressed is point 1.2) Second major revision: quality of the a-zein AF2 predicted 3D model. Considering the tremendous amount of work that has been conducted, my suggestion is to do analyses of the C-ter three a-helix bundle only in addition to those already obtained on the full-length and discuss further the results in light of the general points raised out by this Reviewer on the model quality.

This manuscript is of interest for this Journal and the scientific community, as Reviewer #2 pointed out, it is a comprehensive study describing the landscape of *possible* structures for further investigation.

We look forward to receiving your revised manuscript.

Kind regards,

Matteo De March

Academic Editor

PLOS ONE

Journal Requirements:

5. Review Comments to the Author

Reviewer #1: In this work, the Author characterized the structural ensemble of alpha-zein cZ19C2 using a model predicted by AlphaFold2 (AF2), followed by all-atom and coarse-grained MD simulations on the model. Started with a loosely packed initial AF2 model, the protein exhibited high conformational heterogeneity in simulations, suggesting that ensemble-based approaches are preferred over single native-structure based ones. Valuable observations were obtained – including expanded conformations in high ethanol conditions.

Overall, this manuscript presented substantial and valuable results in many aspects. There are, however, several points that require attention:

Author: I thank the reviewer for feedback on the manuscript. 

Reviewer: 1. The MD simulations suggest the C-terminal helical bundle has a relatively stable structure compared to the rest of the protein, Figures 14A and B may indicate a disordered N-terminal region and a more stable C-terminal core region. Also, sequence from segment 6 to C-terminal end agrees well with the consensus repeat (Figure 3).

In light of these, I wonder if AF2 prediction on the shortened sequence (either from segment 6 to C-terminus or helices 4-6) will yield a structure with better confidence. If yes, it may be interesting to model these regions separately before their integration, perhaps in future studies.

Author: I agree with the reviewer that it is interesting to investigate AF2 predictions on partial C-terminal sequences, i.e., from segment 6 to the C-terminus and helices 4 - 6. Hence, I have carried out additional AF2 predictions for both of these partial sequences. The results are presented and discussed in a new appendix for the revised manuscript, named: S4 Appendix. Comparison of full and partial AlphaFold2 α-zein models. To summarize, when each of these regions was modeled independently, higher local model confidence estimates (pLDDT) were observed. However, upon superimposing the models, the structure of the C-terminal α-helical bundle remained highly similar across all three models. This indicates robustness, or internal consistency, in the AF2 predictions. 

The following text has been added to the revised manuscript, Line 409:

“To further examine the apparently more structurally well-defined C-terminal part of the protein, the full-sequence AlphaFold2 model was compared with partial-sequence AlphaFold2 models built from N89-F219 (segments 6 to 12 in Fig 3A) and N144-T203 (the C-terminal three-helix bundle). As shown in S4 Appendix, structural superposition revealed the α-helical bundle to be highly similar across the three models, indicating internal consistency in the AlphaFold2 predictions.”

Reviewer: 2. What is the rationale of using OPLS forcefield? Does it provide a more accurate modelling on proteins with disordered regions such as the one studied here?

Author: The main rationale was to use a forcefield that strikes a balance between generality and accuracy in representing protein structures and dynamics. OPLS-AA/L emerged as a suitable choice. While OPLS-AA/L is not specifically tailored for disordered regions, its long history of use to study diverse biomolecular systems made it a pragmatic choice for investigating proteins with varying structural characteristics (such as this AlphaFold2 model). Reference simulations at 100 mol% water and 24 mol% ethanol concentrations were studied with the newer field ff99SB*-ILDN, to serve as a supplementary evaluation of certain trends observed in the OPLS-AA/L simulation series.

Reviewer: 3. It may be interesting to follow and elucidate conformational changes and key molecular interactions near the point of divergence in different replicates at early time points in the simulation. Besides Figure 14, a detailed comparison at early time points between replicates, in various conditions, will enhance the manuscript.

Author: I agree that inspection of early structural divergence between different simulation replicates may be interesting. To address this, I have included a new appendix with the revision examining this aspect, titled “S5 Appendix. Early structural divergence across simulation replicates (seeds)”. In summary the analysis shows that structural divergence is substantial already early in the simulations (before 20 ns) consistent with the loose packing of the AlphaFold2 model. The most loosely packed regions of the model tend to undergo the largest displacements, with no obvious preference for the direction of structural deformation. These findings are expected and in line with the results previously discussed in the main text. The following has been added to the main text of the revised manuscript at Line 523:

“The divergence between trajectories differing by seed was substantial already early in the simulations, see S5 Appendix.”

Reviewer: Minor points concerned with general layout and illustrations:

4. In my opinion, the Abstract contains a lot of technical quantitative descriptions, that would be better replaced with more concise descriptions on major findings and their implications, in order to attract readers.

Author: In response to the reviewer's suggestions, the abstract has been modified in the revised manuscript. The updated abstract is presented below:

”α-zeins are amphiphilic maize seed storage proteins with material properties suitable for a multitude of applications e.g., in renewable plastics, foods, therapeutics and additive manufacturing (3D-printing). To exploit their full potential, molecular-level insights are essential. The difficulties in experimental atomic-resolution characterization of α-zeins have resulted in a diversity of published molecular models. However, deep-learning α-zein models are largely unexplored. Therefore, this work studies an AlphaFold2 (AF2) model of a highly expressed α-zein using molecular dynamics (MD) simulations. The sequence of the α-zein cZ19C2 gave a loosely packed AF2 model with 7 α-helical segments connected by turns/loops. Compact tertiary structure was limited to a C-terminal bundle of three α-helices, each showing notable agreement with a published consensus sequence. Aiming to chart possible α-zein conformations in practically relevant solvents, rather than the native solid-state, the AF2 model was subjected to MD simulations in water/ethanol mixtures with varying ethanol concentrations. Despite giving structurally diverse endpoints, the simulations showed several patterns: In water and low ethanol concentrations, the model rapidly formed compact globular structures, largely preserving the C-terminal bundle. At ≥ 50 mol% ethanol, extended conformations prevailed, consistent with previous SAXS studies. Tertiary structure was partially stabilized in water and low ethanol concentrations, but was disrupted in ≥ 50 mol% ethanol. Aggregated results indicated minor increases in helicity with ethanol concentration. β-sheet content was consistently low (~1%) across all conditions. Beyond structural dynamics, the rapid formation of branched α-zein aggregates in aqueous environments was highlighted. Furthermore, aqueous simulations revealed favorable interactions between the protein and the crosslinking agent glycidyl methacrylate (GMA). The proximity of GMA epoxide carbons and side chain hydroxyl oxygens simultaneously suggested accessible reactive sites in compact α-zein conformations and pre-reaction geometries for methacrylation. The findings may assist in expanding the applications of these technologically significant proteins, e.g., by guiding chemical modifications.”

Reviewer: 5. There are many brief sections in Methods that should be merged. For example: “GROMACS All-atom Simulations” & “GROMACS All-atom Simulations Protocol”.

Author: In response to this point, I have removed headings for the following brief sections in Methods: “All-atom Simulations with the OPLS-AA/L Force Field”, “GROMACS all-atom simulation protocol”, and “All-atom Simulations with the ff99SB*-ILDN Force Field”.

Reviewer: 6. Typological error at line 318 “…the sequence bas been…”

Author: I have corrected this error in the revised manuscript.

Reviewer: 7. Similar to 5, the Result part of the manuscript would benefit from merging related sections, if appropriate. In general, the readability could be better improved with a more condensed layout and descriptive section titles.

Author: I have merged related parts of the Results section when deemed appropriate in the revision. For example, the following headings for brief sections have been removed: “Sequence”, “Initial Scan for Self-assembly Domains”, “AlphaFold2 Model”. The content has been merged in the new section titled “α-zein Sequence and Initial Structural Model” with minor adjustments to the relevant text. Furthermore, I have introduced an additional heading level in order to reorganize several subsections, e.g., under “400 ns All-atom MD Simulations”, aiming to provide a more condensed layout. All changes can be tracked in the revised manuscript. 

Reviewer: 8. In my opinion, some main figures could be combined to reduce redundancy, for example Figures 4-8. There are also too many Supplementary Figures (36), this number should be reduced

Author: I acknowledge the reviewer’s opinion about reducing the number of figures, but I would like to note that my use of figures aligns with the journal's liberal figure policy i.e.:

“There are no restrictions on word count, number of figures, or amount of supporting information.” (https://journals.plos.org/plosone/s/submission-guidelines). 

The number of figures can only be reduced by (a) excluding figures or (b) by merging multiple figures. I have already considered both approaches while preparing the manuscript. For instance, the reviewer suggests merging Figures 4 – 8, which already consist of subfigures. Further merging of Figures 4 – 8 would lead to a very large figure, even after excluding certain elements that could be considered redundant. The resolution of such a figure would be lower relative to the current individual figures, since the journals “Figure File Requirements” states a maximum figure file size of 10 MB. In my opinion, sacrificing figure resolution for space is not a satisfactory tradeoff here. Similarly, omitting figures solely to condense the manuscript seems to contradict the relatively free-format ethos of the journal. Given these considerations, I have decided not to reduce the number of figures. 

Reviewer: 9. In Figure 11, the uniform orange “bars” can be confusing, they may be better represented by other means.

Author: I agree that there is (always) room for improving graphical representations. However, the current one represents the best effort on my part, and I believe it provides a serviceable depiction of helicity versus sequence. The reviewer has not offered a specific explanation for finding the orange bars potentially confusing. This leads me to speculate that the figure's description could be the culprit. Therefore, I have revised the Fig 11 caption using the alternative formulation below to improve clarity:

“The height of the blue bars in the foreground indicates the helicity in percent from MD simulations (sum of 3-10 and α-helicity of the last 100 ns of each 400 ns simulation averaged over all 15 × 400 ns MD simulations.) The uniform orange bars in the background indicate positions of α-helical residues in the AlphaFold2 model.”

Reviewer: 10. To help visual comparison of these highly heterogenous structures in most figures, coloring the ribbon presentation of protein structures by residue numbering (such as S2 Appendix C) may further enhance the quality of these illustrations.

Author: I appreciate and share the reviewer’s focus on molecular visualization. Depiction of 3D protein structures in 2D is always challenging and further complicated by differing opinions on the most effective mode of repre

---

## [Decision Letter · Decision Letter 1]

15 Apr 2024

Conformations of a highly expressed Z19 α-zein studied with AlphaFold2 and MD simulations

PONE-D-23-34203R1

Dear Prof. Niels Johan Christensen,

We’re pleased to inform you that your manuscript has been judged scientifically suitable for publication and will be formally accepted for publication once it meets all outstanding technical requirements.

Kind regards,

Matteo De March 

Academic Editor

PLOS ONE

Joseph P.R.O. Orgel

Section Editor

PLOS ONE

Reviewer's Responses to Questions

**Comments to the Author**

1. If the authors have adequately addressed your comments raised in a previous round of review and you feel that this manuscript is now acceptable for publication, you may indicate that here to bypass the “Comments to the Author” section, enter your conflict of interest statement in the “Confidential to Editor” section, and submit your "Accept" recommendation.

Reviewer #1: All comments have been addressed

Reviewer #3: (No Response)

2. Is the manuscript technically sound, and do the data support the conclusions?

Reviewer #1: Yes

Reviewer #3: No

3. Has the statistical analysis been performed appropriately and rigorously? 

Reviewer #1: Yes

Reviewer #3: No

4. Have the authors made all data underlying the findings in their manuscript fully available?

Reviewer #1: Yes

Reviewer #3: Yes

5. Is the manuscript presented in an intelligible fashion and written in standard English?

Reviewer #1: Yes

Reviewer #3: Yes

6. Review Comments to the Author

Reviewer #1: (No Response)

Reviewer #3: On my many comments, the author repeatedly says "it was not the scope of the article" which was frustrating while I spent a lot of time to read the manuscript. On the predicted structure, I appreciated he added the a-zein per-residue pLDDT scores and PAE matrix that I requested.

7. PLOS authors have the option to publish the peer review history of their article (what does this mean?). If published, this will include your full peer review and any attached files.

Reviewer #1: No

Reviewer #3: No
